# Gaia2: Benchmarking LLM Agents on Dynamic and Asynchronous Environments

**Romain Froger, Pierre Andrews, Matteo Bettini, Amar Budhiraja, Ricardo Silveira Cabral, Virginie Do, Emilien Garreau, Jean-Baptiste Gaya, Hugo Laurençon, Maxime Lecanu, Kunal Malkan, Dheeraj Mekala, Pierre Ménard, Gerard Moreno-Torres Bertran, Ulyana Piterbarg, Mikhail Plekhanov, Mathieu Rita, Andrey Rusakov, Vladislav Vorotilov, Mengjue Wang, Ian Yu, Amine Benhalloum, Grégoire Mialon, Thomas Scialom**

Meta SuperIntelligence Labs
{rfroger, amineben, gmialon, tscialom}@meta.com

## Abstract

We introduce **Gaia2**, a benchmark for evaluating large language model agents in realistic, asynchronous environments. Unlike prior static or synchronous evaluations, Gaia2 introduces scenarios where environments evolve independently of agent actions, requiring agents to operate under temporal constraints, adapt to noisy and dynamic events, resolve ambiguity, and collaborate with other agents. Each scenario is paired with a write-action verifier, enabling fine-grained, action-level evaluation and making Gaia2 directly usable for reinforcement learning from verifiable rewards. Our evaluation of state-of-the-art proprietary and open-source models shows that no model dominates across capabilities: GPT-5 (high) reaches the strongest overall score of 42% pass@1 but fails on time-sensitive tasks, Claude-4 Sonnet trades accuracy and speed for cost, Kimi-K2 leads among open-source models with 21% pass@1. These results highlight fundamental trade-offs between reasoning, efficiency, robustness, and expose challenges in closing the "sim2real" gap. Gaia2 is built on a consumer environment with the open-source **Agents Research Environments** platform and designed to be easy to extend. By releasing Gaia2 alongside the foundational ARE framework, we aim to provide the community with a flexible infrastructure for developing, benchmarking, and training the next generation of practical agent systems.

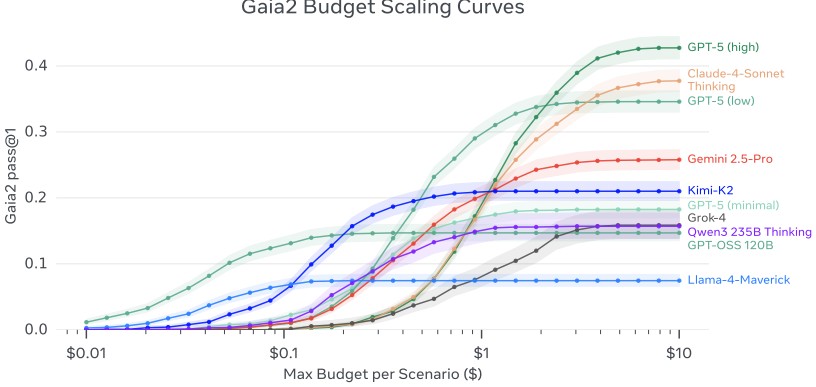

Figure 1: Gaia2 budget scaling curve: for each max_budget, we plot $\sum \mathbb{1}\{\text{scenario\_result} = \text{True} \wedge \text{scenario\_cost} < \text{max\_budget}\}$. Equipped with a simple ReAct-like scaffold (see Section 3), no model evaluated here dominates across the intelligence spectrum—each trades off capability, efficiency, and budget. At equal cost, some models fare better, yet all curves plateau, suggesting that standard scaffolds and/or models miss ingredients for sustained progress. Cost estimates from Artificial Analysis model pricing data (accessed September 10, 2025).

## 1 INTRODUCTION

Reinforcement learning from verifiable rewards (RLVR) has emerged as a promising path for improving large language model (LLM) agents at scale in domains such as reasoning, coding, and tool-use, offering a more reliable alternative to preference-based methods (OpenAI, 2024b; DeepSeek-AI et al., 2025; Mistral-AI et al., 2025; MoonshotAI et al., 2025). At the same time, the use-cases of modern agents increasingly involve sustained long-horizon interaction with dynamic environments, where time, uncertainty, and collaboration play a central role. This has motivated the creation of LLM agent benchmarks (Mialon et al., 2023; Jimenez et al., 2024; Yao et al., 2024; Backlund & Petersson, 2025), yet most such benchmarks are static or synchronous: environments only change when the agents act, and evaluation typically ignores intermediate steps or actions. As a result, many of the challenges agents face in real deployments—such as handling asynchronous events, operating under temporal constraints, or adapting to noise and uncertainty—remain untested.

We introduce **Gaia2**, a benchmark designed to address these limitations by evaluating agents in asynchronous environments with verifiable tasks that, like GAIA (Mialon et al., 2023), are simple for humans but challenging for today's AI models. Gaia2 scenarios are motivated by real deployed use cases: it generalizes information seeking to environments instead of web-only, Gaia2-Time reflects the requirements of scheduled task products (e.g., calendar and reminders), and Gaia2-Agent2Agent mirrors the recently proposed Agent2Agent protocol for interoperable multi-agent systems (Google Developers, 2025). Gaia2 consists of 1,120 human-annotated scenarios set in a smartphone-like environment with realistic apps (email, messaging, calendar, contacts, etc.), similar to AppWorld and ToolSandbox (Trivedi et al., 2024; Lu et al., 2024). Each scenario requires capabilities beyond search and execution, including adaptability to new events, robustness to noise, resolution of ambiguity, temporal awareness, and collaboration with other agents. To enable reproducible and fine-grained evaluation, Gaia2 introduces a `write` action verifier that checks every state-changing action against oracle annotations, making the benchmark directly applicable to RLVR. Built on the **Agents Research Environments** (ARE) platform, Gaia2 provides abstractions for creating asynchronous environments and supports continuous extension of benchmarks. The core concepts of ARE, illustrated in Figure 2, allow generalization beyond Gaia2 to the definition of other benchmarks. In practice, this design reveals new failure modes: while frontier models achieve overall success rates around 42%, no system dominates across all capabilities, with strong reasoning often traded off against speed, robustness, or cost.

**Contributions** This paper makes three main contributions to advance the evaluation of LLM agents and to chart open directions for the next generation of practical systems:

- **ARE framework:** We release *Agents Research Environments*, a general-purpose platform for building asynchronous, event-driven benchmarks that support scalable evaluation and data generation for RL.
- **Gaia2 benchmark:** We introduce *Gaia2*, the first benchmark unifying asynchronous execution, temporal reasoning, noise robustness, ambiguity resolution, and multi-agent collaboration under a verifiable evaluation framework directly usable for RLVR.
- **Empirical study:** We evaluate leading proprietary and open-source models on Gaia2, exposing fundamental trade-offs between reasoning strength, efficiency, robustness, and cost.

## 2 RELATED WORK

**Benchmarking LLM agents** A wide range of benchmarks have been proposed to measure agent capabilities. Embodied and web-based environments such as ALFWorld (Shridhar et al., 2021), WebShop (Yao et al., 2023a), WebArena (Zhou et al., 2024), and WorkArena (Drouin et al., 2024) emphasize grounded execution. Synthetic environments such as AppWorld (Trivedi et al., 2024) and ToolSandbox (Lu et al., 2025) introduce app-like tasks with state verification or milestone-based evaluation, while BFCL (Patil et al., 2025) targets large-scale function calling. Other efforts incorporate temporal dynamics and multi-agent interaction, including VendingBench (Backlund & Petersson, 2025), $\tau$-Bench and $\tau^2$-Bench (Yao et al., 2024; Barres et al., 2025), MultiAgentBench (Zhu et al., 2025), and MCP-based benchmarks (Wang et al., 2025; Team, 2025; Gao et al., 2025; Anthropic, 2024). Finally, static setups such as GAIA (Mialon et al., 2023), SWE-bench (Jimenez

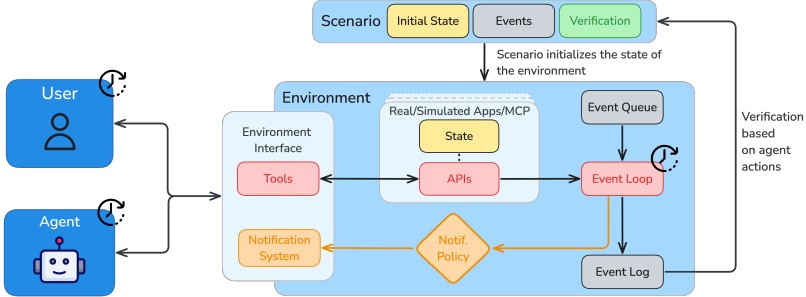

Figure 2: ARE environments are event-based, time-driven simulations, that run asynchronously from the agent and the user. ARE environments allows playing scenarios, which typically contain tasks for the agent and verification logic. Whether initiated by agent or user, interactions happen through the same interfaces and can be either tool calls, or tool output/notification observations. Extensive simulation control and logging allow precise study of agents behavior.

et al., 2024), and BrowseComp (Wei et al., 2025) evaluate only final outcomes. While these benchmarks each capture valuable aspects of agent reasoning, tool use, or collaboration, they remain *synchronous and agent-driven*: environments only change when the agent acts, and evaluation typically ignores intermediate steps or actions. Gaia2 differs by introducing asynchronous, event-driven environments that stress temporal constraints, robustness, ambiguity resolution, and multi-agent coordination under a unified, verifiable evaluation.

**Verification in agentic benchmarks** Verification strategies vary across benchmarks. GAIA (Mialon et al., 2023) evaluates correctness at the final output level via exact match. This suits search-style tasks but lacks flexibility in format and content, especially in web-based, evolving domains. ToolSandbox (Lu et al., 2025) introduces *milestones* and *minefields* that constrain the agent's trajectory, enabling early checks of both outcomes and intermediate behavior. Beyond strictly verifiable domains, the *Rubrics as Rewards* framework (Gunjal et al., 2025; Starace et al., 2025; Lin et al., 2025) shows how checklist-style rubrics can serve as interpretable reward signals for subjective tasks, highlighting the broader potential of rubric-based evaluation. Gaia2 extends this with the *ARE Verifier*, which evaluates every state-changing write action against oracle annotations. It combines exact argument checks, rubric-guided judgments for flexible cases, and causal and temporal constraints. Importantly, the verifier is a standalone contribution: a general mechanism for fine-grained, reproducible credit assignment reusable beyond Gaia2. While today's models underperform, we expect future RLVR-trained systems to close the gap and eventually solve Gaia2.

## 3 ARE: SCALING UP AGENT ENVIRONMENTS AND EVALUATIONS

ARE is a research platform for creating simulated environments, running agents in them, and analyzing their behavior. ARE environments evolve continuously and are decoupled from the agent. Time advances in the simulation as the environment introduces events. Agents run asynchronously and interact with the user and environment through dedicated interfaces.

**Core concepts** At its foundation, ARE introduces a set of abstractions, illustrated in Figure 2, that make it possible to design rich, dynamic environments. More precisely: (i) *apps* are stateful APIs with associated content, analogous to applications such as messaging or email, each exposing tools that can be typed as read-only or write, enabling fine-grained control and verification; (ii) a collection of apps together with a time manager and governing rules forms an *environment*, which can host one or several agents; (iii) within these environments, *events* represent everything that happens, from tool calls and state changes to scheduled updates, and are fully logged, scheduled either at absolute timestamps or relative to others, and organized into dependency graphs; (iv) to surface relevant dynamics, *notifi-*

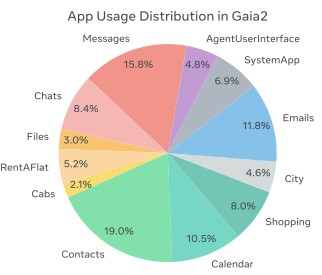

Figure 3: App usage distribution across the 12 `Mobile` apps in Gaia2 for Llama 4 Maverick.

*cations* provide a configurable observability layer: a policy selects which events are pushed to the agent's context, enabling the study of proactive and reactive behavior under varying observability; and (v) *scenarios* extend static tasks into dynamic trajectories by specifying an initial state and a DAG of events, including the user's request, intermediate events, and a verification method. Verification can run offline at the end of the run or online via scheduled validation events, and focuses on write operations to avoid over-constraining exploration strategies. To demonstrate the generality of these abstractions, we validated that ARE can faithfully reimplement existing agentic benchmarks such as $\tau$-bench, $\tau^2$-bench GAIA, and BFCL-v3, VendingBench(Yao et al., 2024; Barres et al., 2025; Mialon et al., 2023; Patil et al., 2025; Backlund & Petersson, 2025), confirming that the platform both subsumes current benchmarks and provides a foundation for the next generation of agentic evaluations. More details about ARE concepts are provided in Appendix A.1.

**Asynchronicity and time**   Because environments run asynchronously, model generations directly consume simulated time: if an agent takes longer to respond, the environment clock still advances, and external events may happen during its reasoning process. This design unlocks evaluations of temporal awareness and responsiveness, which are impossible to capture in synchronous settings.

**Mobile environment**   To demonstrate the versatility of the ARE abstractions, we release `Mobile` as an instantiation of a consumer mobile environment. It features twelve apps (Messages, Chats, Emails, Calendar, Contacts, Shopping, Cabs, Files, etc.) and 101 associated tools, similar in spirit to AppWorld (Trivedi et al., 2024) and ToolSandbox (Lu et al., 2024). Each *"universe"* represents a complete instance of this environment—the full state of all apps centered around a specific user. Applications are populated with synthetic but coherent data, seeded with personas sampled from PersonaHub (Ge et al., 2024) and propagated across apps via a dependency graph to ensure cross-app consistency (e.g., contacts align across messaging and email, events match calendar availability). Universes contain between 400K and 800K tokens of structured and unstructured content (excluding filesystem contents), making them suited for long-context and long-horizon tasks. `Mobile` is governed by clear rules: each turn starts with a user message or a notified event and ends when the agent replies to the user. During the turn, simulated time advances continuously, and scenarios terminate either on task completion, when constraints on time or steps are exceeded, or when verification fails. While `Mobile` focuses on the consumer domain to leverage a unified app concept, the underlying ARE platform is environment-agnostic. The API definitions remain invariant across domains—for example, the interface for a *Chats* tool is identical whether in a mobile or desktop setting. Consequently, the architecture presented here extends naturally to other domains such as desktop automation, customer support, and web browsing, where creating a new environment requires only defining the relevant tool interfaces.

**Agent orchestration**   Running agents in ARE requires an orchestration compatible with its abstractions. For a fair evaluation, we use a model-agnostic scaffold based on a ReAct loop (Yao et al., 2023b), where the agent outputs one tool call per step in structured JSON. The orchestration is augmented with `pre-step` and `post-step` hooks: before each LLM call, notifications queued in the environment are injected into the agent's context; after the tool call, the agent termination condition is checked. This minimal extension preserves the simplicity of ReAct while making it compatible with asynchronous and multi-turn environments. Alternative orchestrations can be easily plugged in. To ensure that this sequential scaffolding does not artificially bottleneck performance, we compared it against a Parallel Tool Calling (PTC) orchestration in Appendix B.3.2. Results show that PTC can improve efficiency (wall clock time and token usage) but not performance (see Table 6), confirming that the observed limitations are intrinsic to model capabilities rather than the scaffold.

## 4   GAIA2: EXPANDING GENERAL AGENT EVALUATION

Building on the abstractions of ARE, we introduce **Gaia2**, consisting of 800 unique verifiable scenarios, carefully annotated by humans across 10 distinct universes in the `Mobile` environment, with 101 tools each. The scenarios are organized into splits, each targeting one agent capability defined below. To support rapid and cost-effective evaluations, we also curate a 160-scenario subset, Gaia2-mini. The benchmark includes two augmentation setups derived from Gaia2-mini, adding 320 scenarios to the original 800 for a total of 1,120 scenarios.

| Capability | Example Task | Explanation |
|---|---|---|
| Execution | *Update all my contacts aged 24 or younger to be one year older than they are currently* | Evaluates the ability to chain long seq. of `write` **actions** in the right order |
| Search | *Which city do most of my friends live in? In case of a tie, return the first city alphabetically* | Evaluates the ability to chain long seq. of **read actions** in the right order |
| Ambiguity | *Schedule a 1h Yoga event each day at 6:00 PM from October 16, 2024 to October 21, 2024* | Tests whether agents ask for **clarification** on impossible, contradictory, or ambiguous tasks |
| Adaptability | *I have to meet my friend Kaida to view a property [...] If she replies to suggest another property or time, update the calendar event* | Requires agents to **adapt dynamical**ly to environmental changes |
| Time | *Send messages to each of the colleagues I am supposed to meet today, asking who is supposed to order the cab. If after 3 minutes there is no response, order a cab from [...]* | Evaluates whether agents can **complete tasks in due time** & maintain temporal awareness |
| Agent2Agent | *\*Same Search task as above but the Contacts and Chats apps are replaced by app sub-agents\** | Tests whether agents can **collaborate with other agents** to use tools & complete tasks |
| Noise | *\*Same Adaptability task as above but with random tool execution errors and random environment events occurring during execution\** | Evaluates whether agents are **robust to environment noise & distractors** |

Figure 4: The seven core agent capabilities evaluated by the splits of Gaia2.

## 4.1 Capabilities evaluated

Gaia2 evaluates agents across 1,120 scenarios. To provide a clear taxonomy, we distinguish between **Core Capabilities** (*Execution, Search, Ambiguity, Adaptability, Time*) and **Augmentations** (*Noise, A2A*). The five core splits comprise 800 unique, human-authored scenarios, each instantiated with a unique event DAG and initial environment state. We treat these core categories as dominant "flavors" rather than strictly orthogonal dimensions. In practice, any natural task is inherently compositional (e.g., a *Time* task often requires *Search* and *Execution*). Consequently, we explicitly chose not to introduce a separate "compositional" split; our early experiments with scenarios artificially combining three or more distinct capabilities resulted in unnatural tasks that lacked a clear evaluation signal. Instead, we rely on the organic compositionality present in the core splits to ensure tasks remain realistic while still allowing for clear failure-mode attribution.

**Environment augmentations**   The *Noise* and *Agent-to-Agent (A2A)* splits are environment-level modifiers applied to base scenarios to stress-test robustness and collaboration. Because our verifier checks state changes rather than specific tool traces, these augmentations do not require new annotations. In the **Noise** split, we inject controlled perturbations, including tool anomalies (e.g., random execution failures, signature changes) and irrelevant environment events (e.g., incoming spam emails). In the **A2A** split, apps are replaced by "app-agents". The main agent loses direct access to these apps' tools and must instead coordinate with the app-agents via messaging to solve the task. App-agents are not fully autonomous: they are invoked on-demand to execute specific subtasks and return a report. This setting explicitly evaluates the main agent's ability to decompose goals and coordinate under partial observability. In our evaluation setting, main- and app-agents use the same underlying model.

## 4.2 Scenario design and annotation protocol

We construct Gaia2 scenarios using the ARE annotation interface (see Appendix A.4 for details), which lets annotators explore a generated `Mobile` universe. Their task is to create DAGs of `write` actions and environment events as ground truth. Starting from the generated environment, annotators design scenarios that isolate and stress a single capability at a time (e.g., Adaptability, Time), ensuring a clear signal of model strengths and weaknesses.

Each scenario undergoes multiple rounds of validation by independent annotators, followed by a consistency check. We complement this process with automated guardrails (e.g., structural constraints on event graphs) and post-hoc difficulty calibration using a baseline agent. This combination yields a diverse, challenging, and verifiable set of scenarios while reducing annotation errors. We provide more details on our annotation process and guidelines in Appendix B.1.

Table 1: ARE Verifier and In-context Verifier on 450 hand-labeled validation trajectories.

| Verifier | Agreement | Precision | Recall |
|---|---|---|---|
| In-context Verifier (LLM judge only) | 0.72 | 0.53 | 0.83 |
| ARE Verifier | 0.98 | 0.99 | 0.95 |

### 4.3 VERIFIER

The ARE Verifier evaluates agent trajectories against a minimal oracle sequence. Unlike final-answer judges, it is *goal-oriented* rather than *path-optimal*. We distinguish `read` from `write` actions: agents may execute unlimited `read` steps (e.g., browsing) for exploration without penalty, as only `write` steps modify the environment and count toward goals. The verifier is also order-agnostic regarding independent goals.

Evaluation covers four dimensions: (i) **Consistency**—tool names/counts must match; arguments use exact matching for rigid fields (IDs) and LLM rubrics for flexible ones (text), with anti-hacking checks; (ii) **Causality**—actions must respect dependency DAGs (parents before children); (iii) **Timing**—enforced via tolerance windows; and (iv) **Completeness**—success requires matching all oracle `write` actions.

On 450 labeled trajectories (Table 1), the verifier achieves 0.98 agreement and 0.99 precision, outperforming LLM-only baselines. Beyond Gaia2, it serves as a reusable component for faithful benchmarking and RLVR training in any ARE-based environment. Further details are in Appendix B.2.

## 5 EXPERIMENTS

In our core experiments, we evaluate state-of-the-art models on each Gaia2 capability split (MoonshotAI et al., 2025; Gemini Team, 2025; Yang et al., 2025; Llama Team, 2024; OpenAI, 2024a). We also test the sensitivity of models to various evaluation configurations for Time and Agent2Agent.

**Experimental setup** We use the same ReAct-style baseline scaffold (Section 3) for all evaluations in order to ensure consistent comparisons across models and providers. All LLMs are evaluated at full context length ($\geq$128K tokens), temperature 0.5, and 16K token generation limits per turn. Scenarios are run three times to account for potential variance, and are terminated when one of the following conditions is met:

(i) 200 steps, (ii) context overflow, i.e., the agent exceeds the available context window (failure), (iii) verification completion, i.e., the verifier determines the trajectory outcome—either by failing at some turn or by successfully passing verification at every turn, or (iv) timeout. The environment provides tools and notifications via system prompts, with notification verbosity set to `medium` by default: agents receive systematic alerts for high-priority events while filtering out lower-priority background notifications.

We handle deployment issues like outages and rate limits using a `simulated` generation time—pausing during responses and resuming with a matching time offset—to preserve realistic timing while enabling robust evaluation. The ARE Verifier uses `Llama-3.3-70B-Instruct` at temperature 0. For more details on our experimental procedure, please see Appendix B.4.

### 5.1 CORE RESULTS

Our core experimental results are presented in Table 2, Figure 5, and Figure 6. Among Gaia2 splits, *Execution* and *Search* emerge as the easiest, consistent with prior benchmark saturation (Trivedi et al., 2024; Lu et al., 2024). *Ambiguity* and *Adaptability* remain challenging, with only Claude-4-Sonnet and GPT-5 (high) achieving robust performance. The *Time* split further differentiates frontier models: only Gemini 2.5 Pro and Sonnet achieve meaningful scores, reflecting their efficiency-latency advantages (Figure 6). *Noise* robustness also lags, with most models scoring below 20 despite GPT-5 (high) reaching 35.4%. *Agent2Agent* collaboration benefits weaker models more than frontier systems (see Figure 10). Overall, GPT-5 (high) leads with 42.1% pass@1, maintaining an 8-point margin over Sonnet across all categories. Kimi-K2 distinguishes itself among open mod-

Table 2: Pass@1 scores on Gaia2 scenarios per model and capability split. All models are evaluated with the same baseline ReAct scaffolding described in Section 3 and with three runs to account for potential variance. The overall score is the average across splits.

| | Execution | Search | Ambiguity | Adaptability | Time | Noise | Agent2Agent | Overall |
|---|---|---|---|---|---|---|---|---|
| Llama 3.3 70B Instruct | 7.1 ±1.2 | 11.5 ±1.5 | 1.7 ±0.6 | 1.9 ±0.6 | 0.4 ±0.3 | 3.8 ±0.9 | 4.6 ±1.0 | 4.4 |
| Llama 4 Maverick | 13.8 ±1.6 | 14.4 ±1.6 | 2.1 ±0.7 | 5.0 ±1.0 | 1.2 ±0.5 | 6.2 ±1.1 | 9.2 ±1.3 | 7.4 |
| GPT-4o | 8.3 ±1.3 | 17.5 ±1.7 | 4.4 ±0.9 | 6.2 ±1.1 | 5.8 ±1.1 | 4.6 ±1.0 | 5.2 ±1.0 | 7.4 |
| GPT-OSS 120B (high) | 17.9 ±1.8 | 33.1 ±2.1 | 8.3 ±1.3 | 10.6 ±1.4 | 0.6 ±0.4 | 14.6 ±1.6 | 10.6 ±1.4 | 13.7 |
| Qwen3-235B | 22.7 ±1.9 | 22.3 ±1.9 | 6.5 ±1.1 | 8.1 ±1.2 | 1.2 ±0.5 | 10.8 ±1.4 | 9.4 ±1.3 | 11.6 |
| Qwen3-235B Thinking | 28.1 ±2.1 | 36.2 ±3.8 | 10.0 ±2.4 | 16.2 ±2.9 | 0.0 ±0.0 | 6.9 ±2.0 | 12.5 ±2.6 | 15.7 |
| Grok-4 | 8.8 ±2.2 | 57.5 ±3.9 | 9.4 ±2.3 | 4.4 ±1.6 | 0.0 ±0.0 | 15.6 ±2.9 | 14.4 ±2.8 | 15.7 |
| Kimi-K2 | 34.2 ±2.2 | 36.0 ±2.2 | 8.3 ±1.3 | 24.0 ±1.9 | 0.8 ±0.4 | 18.8 ±1.8 | 18.3 ±1.8 | 20.1 |
| Gemini-2.5-Pro | 39.2 ±2.2 | 57.7 ±2.3 | 17.5 ±1.7 | 17.5 ±1.7 | 7.3 ±1.2 | 20.4 ±1.8 | 20.4 ±1.8 | 25.8 |
| Claude-4-Sonnet | 57.9 ±2.3 | 59.8 ±2.2 | 24.2 ±2.0 | 38.1 ±2.2 | 8.1 ±1.2 | 27.7 ±2.0 | 27.9 ±2.0 | 34.8 |
| Claude-4-Sonnet Thinking | 62.1 ±2.2 | 60.6 ±2.2 | 27.3 ±2.0 | 42.1 ±2.3 | 8.5 ±1.3 | 31.2 ±2.1 | 32.5 ±2.1 | 37.8 |
| GPT-5 (minimal) | 31.9 ±2.1 | 26.2 ±2.0 | 20.6 ±1.8 | 19.2 ±1.8 | 5.2 ±1.0 | 13.1 ±1.5 | 11.5 ±1.5 | 18.2 |
| GPT-5 (low) | 52.7 ±2.3 | 64.2 ±2.2 | 39.6 ±2.2 | 30.2 ±2.1 | 2.3 ±0.7 | 28.3 ±2.1 | 24.6 ±2.0 | 34.6 |
| GPT-5 (high) | **69.2** ±2.1 | **79.6** ±1.8 | **51.9** ±2.3 | **40.4** ±2.2 | 0.0 ±0.0 | **35.4** ±2.2 | 17.9 ±1.8 | **42.1** |

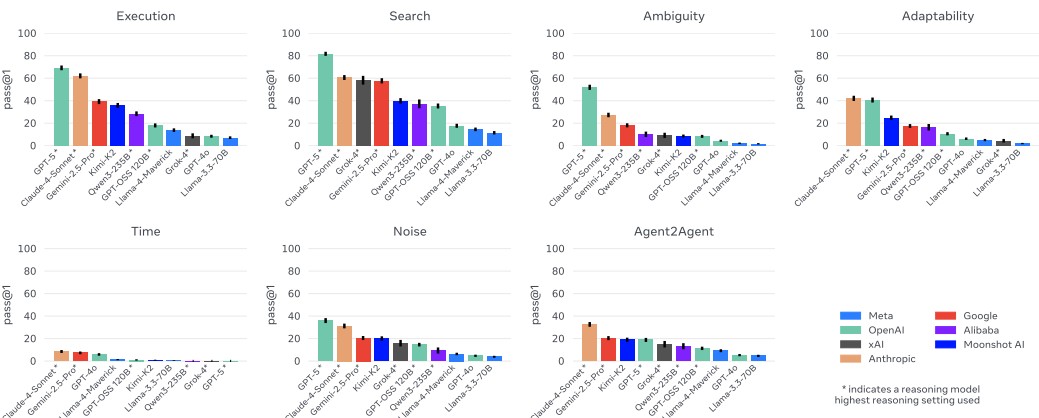

Figure 5: Gaia2 scores per capability split. Models are reranked independently for each capability, highlighting where they excel or struggle.

els, particularly on *Adaptability*. While instruction-following and search tasks are largely solved, robustness, ambiguity resolution, and collaboration remain open challenges.

In Figures 6 and 7, we extend our analysis beyond raw scores to identify the finer-grained factors that drive performance differences between models on Gaia2. In addition, since agents are ultimately intended for deployment in production settings, we evaluate their performance in relation to their computational cost[1] and execution time.

**Cost-performance trade-offs** Figure 6 reveals clear cost-performance-time trade-offs. GPT-5's reasoning models demonstrate direct scaling: higher test-time compute yields better performance but longer solution times. Claude 4 Sonnet costs roughly 3× more than GPT-5 (low) for comparable accuracy but operates much faster. Outliers include the inefficient Grok-4 and cost-effective Kimi-K2. While an average human annotator can solve every task, they are slower than all models, partly due to using ARE's GUI rather than a native OS. These findings highlight the need for cost-normalized evaluation metrics. Comparing model parameters or FLOPs alone inadequately reflects real-world deployment conditions. Success rate per dollar better captures how agents will be judged in practice—by reliable, efficient task completion under resource constraints.

**Performance drivers** We examine behavioral factors correlating with Gaia2 performance. Two hypotheses guide our analysis: (1) exploration drives success through increased tool use and systematic information gathering before write operations, and (2) comprehensive reasoning via token generation improves performance. Figure 7 confirms both relationships: performance correlates

---

[1]Cost estimates from Artificial Analysis model pricing data (accessed September 10, 2025)

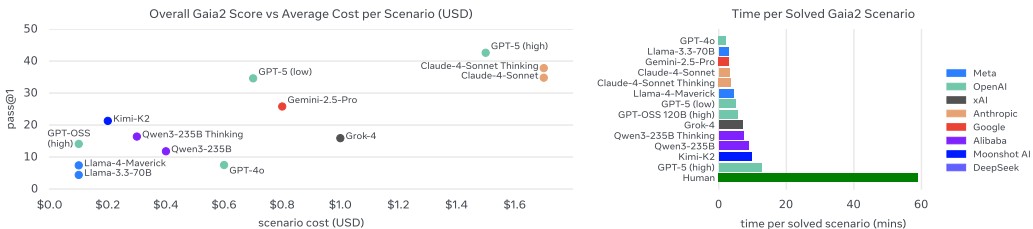

Figure 6: **Left**: Gaia2 score vs average scenario cost in USD. **Right**: Time taken per model to successfully solve Gaia2 scenarios compared to Humans.

positively with tool calls (left) and output tokens (right). However, Claude-4 Sonnet and Kimi-K2 stand out as notable outliers, achieving high performance (35% and 21% respectively) while producing relatively few tokens—suggesting exceptional efficiency, perhaps due to larger parameter counts or specialized architectures. Within families, we observe a striking contrast between the base and "Thinking" variants of Claude and Qwen: the latter generate more tokens per step but take fewer steps overall, leading to higher pass@1 and lower cost per solved scenario, effectively trading verbosity for efficiency (e.g., Qwen-235B Thinking vs. Qwen-235B). App usage patterns were nearly identical across models (Figure 3), indicating that performance differences stem primarily from general reasoning capabilities rather than app-specific preferences.

## 5.2 TIME REVEALS THE IMPACT OF INFERENCE SPEED—AND SYSTEM RELIABILITY

We evaluate Gaia2-Time in two modes. As shown in Figure 8 (left), removing generation latency ("instant" mode) improves all models, with the largest gains for reasoning models: Sonnet rises from 8.1% to 26.7%, and GPT-5 (high) from 0.0% to 34.4%. Weaker models improve modestly due to the difficulty of the tasks, while Gemini 2.5 Pro combines strong performance with low latency and therefore best supports timing requirements. In the default mode, we observe inverse scaling in the *Time* capability: models trade *Time* performance for *Execution* performance due to longer thinking, see Figure 8 right. This underscores the need for adaptive compute—using shallow models and performing deeper reasoning only when necessary. Besides inference speed, the *Time* split also underlines the need for reliable infrastructure to serve responsive models without rate limits and server downtime in order to handle time-sensitive tasks. Finally, some Time scenarios require concurrent actions within narrow windows, which our single-threaded scaffold cannot fully express. Parallel orchestration is a promising direction to solve this type of scenarios.

## 5.3 A CLOSER LOOK AT MULTI-AGENT COLLABORATION ON GAIA2 WITH AGENT2AGENT

Inspired by recent work pushing beyond single-LLM agent tool-use and towards agent teams that message, coordinate, and divide labor (Google Developers, 2025), we study multi-agent collabora-

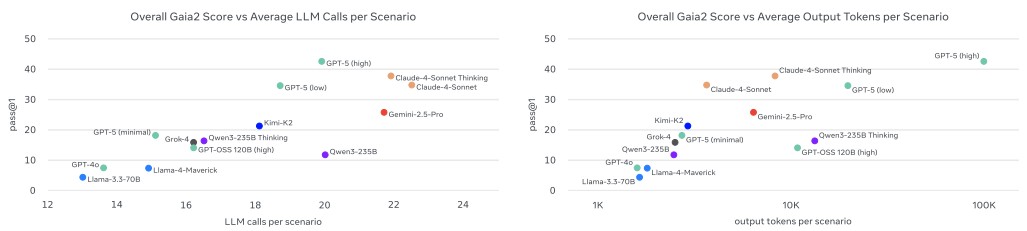

Figure 7: Left: Gaia2 pass@1 versus average model calls per scenario. The performance of models is highly correlated to the number of tool calls, emphasizing the importance of exploration. Right: Gaia2 pass@1 score versus average output tokens per scenario (log scale). Claude 4 Sonnet, while costing a lot exists beyond the Pareto frontier.

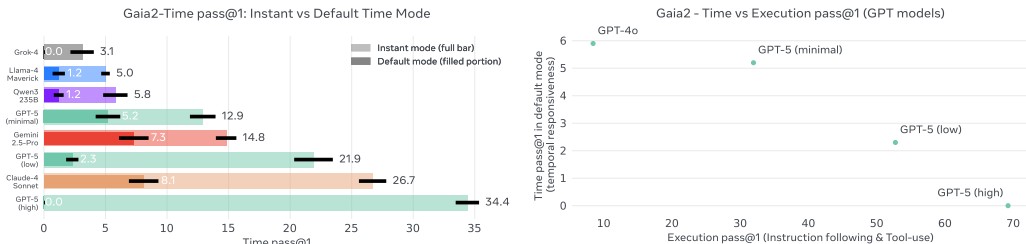

Figure 8: **Left**: Pass@1 on Gaia2-Time in default vs. instant. **Right**: Inverse scaling on Time—reasoning-heavy models are slower and miss deadlines.

tion on Gaia2 scenarios. We focus on two models at different points in the cost-quality curve: Llama 4 Maverick, a lighter-weight model, and Claude 4 Sonnet, the strongest overall LLM on standard Agent2Agent (Table 2).

For the weaker Llama 4 Maverick, centralized collaboration on Gaia tasks improves both performance with pass@k and operational stability. As the agent-to-agent ratio $r$ increases, we observe more favorable scaling with repeated sampling and a lower incidence of tool-call errors per step (Figure 9 right; Figure 10). However, the trends observed for Llama 4 are not universal. For Claude 4 Sonnet, increasing the collaborator ratio $r$ – and thus the degree of task decomposition – does not improve cost-normalized performance under best-of-$k$ sampling: score per token plateaus with or without multi-agent collaboration. Similarly, collaboration ratio with Agent2Agent has a weak negative effect on tool call error frequency.

One explanation for these findings may lie in the fact that Agent2Agent induces hierarchical decomposition into decision-making. As shown in Figure 9 left, sub-goals issued by a main-agent to an app-agent instantiate temporally extended actions akin to options (Sutton et al., 1999). Under this lens, gains in performance may materialize only when the benefits of decomposition outweigh the costs. For example, Agent2Agent may increase task score only when sub-goals set by main-agents are well-scoped and both app- and main-agents are capable of reliably exchanging state & intent during message-passing. Likewise, the addition of hierarchy can result in cascading errors and/or saturating gains if post-training has fit models to long-form, single-agent planning and tool-use; in this regime, coordination may introduce overhead that offsets accuracy and efficiency gains.

Heterogeneous teams open a new compute scaling axis for task automation, for example, by keeping a strong main agent to plan/decompose tasks while swapping in cheaper app-agents to execute sub-goals[2]. Empirically, replacing Llama 4 Maverick app-agents with Claude app-agents boosts pass@1 for both main-agent settings (16.2 with Llama-main, 29.3 with Claude-main), while the fully light

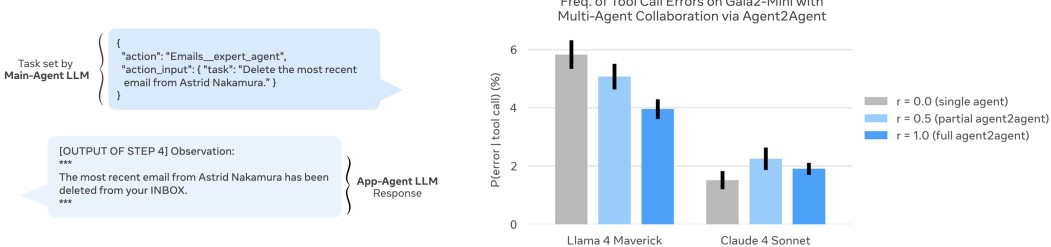

Figure 9: Agent2Agent tests whether LLM agents can collaborate through message passing in order to solve Gaia2 tasks via sub-task decomposition. For lighter-weight LLMs, collaboration in Agent2Agent results in a lower incidence of tool call errors. **Left**: Sample exchange between Llama 4 Maverick main vs app agent in an Agent2Agent scenario. **Right**: Frequency of errors per tool call (lower is better) on Gaia-2 mini for Llama 4 Maverick and Claude 4 Sonnet.

---

[2]ARE natively supports controlled evaluation of heterogeneous teams, making team composition a primary experimental factor alongside standard inference hyperparameters.

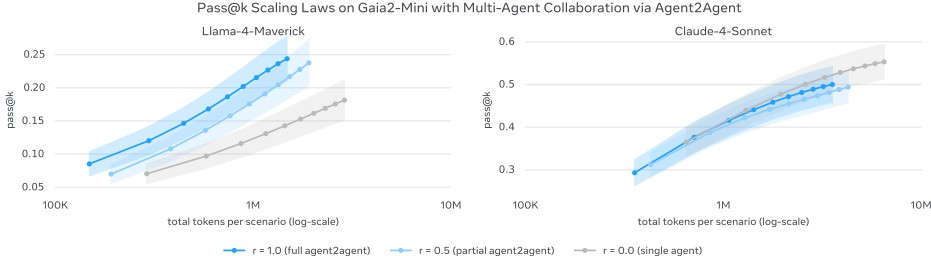

Figure 10: Increasing the number of multi-agent collaborators in Gaia2 scenarios by increasing the Agent2Agent ratio "r" improves pass@k scaling laws for Llama 4 Maverick, but does not improve token cost vs score tradeoffs with repeated sampling for Claude 4 Sonnet.

|  |  | **Main-Agent LLM** | |
|---|---|---|---|
|  |  | Llama 4 Maverick | Claude 4 Sonnet |
| **App-Agent LLM** | Llama 4 Maverick | $8.5 \pm_{1.7}$ | $16.2 \pm_{0.7}$ |
|  | Claude 4 Sonnet | $18.3 \pm_{0.7}$ | $29.3 \pm_{2.9}$ |

Table 3: Probing cross-model collaboration in Gaia2-mini Agent2Agent scenarios: we evaluate pass@1 across main- vs app-agent pairings with Llama 4 Maverick and Claude 4 Sonnet in the fully collaborative Agent2Agent setting ($r = 1$). The results are averaged over three runs and presented with the standard error.

configuration is weakest (8.5). This suggests that for existing LLMs, Gaia2 task completion remains sensitive to execution fidelity at the app-agent level: stronger executors improve outcomes even when the main agent is light. Similarly, pairing a strong main agent with light executors still outperforms the all-light team (18.3 with Claude-main + Llama-app), indicating that higher-quality sub-goal specification and critique from the main-agent contribute independent gains. These findings are consistent with prior work suggesting heterogeneous multi-agent systems can trade planning capacity against execution fidelity to manage compute-quality trade-offs.

## 6 CONCLUSION & DISCUSSION

ARE introduces an asynchronous, event-driven evaluation framework with action-level verification, enabling reproducible benchmarking directly applicable to RLVR. Its abstractions—apps, events, notifications, and scenarios—along with the MOBILE environment provide an extensible foundation for community-driven evaluations and RL data generation. Gaia2 demonstrates that no model dominates across all capabilities: GPT-5 (high) achieves the best overall accuracy (42% pass@1), Claude-4 Sonnet offers competitive performance with lower latency, and Kimi-K2 leads among open-source systems (20%). Scaling curves reveal fundamental cost–time–accuracy trade-offs, highlighting the need for cost-normalized reporting.

Verification at the action level scales more effectively than end-state comparisons and supports fine-grained credit assignment. The ARE Verifier matches human annotations with high fidelity (0.99 precision, 0.95 recall), while uncovering issues such as "judge-hacking." Robust verifier design is thus critical for both evaluation and RL training; hybrid approaches combining scalar rewards with preference signals remain an open direction.

Finally, Gaia2's Time split and A2A experiments underscore the critical role of orchestration. The inverse scaling observed in Time-sensitive tasks suggests that future agents require *adaptive compute* strategies: deploying fast, lightweight reasoning for routine tasks while reserving deeper deliberation for complex ones. Simultaneously, the A2A results demonstrate that orchestration extends to collaboration, where heterogeneous teams can outperform monolithic models through effective delegation.

ACKNOWLEDGEMENTS

We thank Nikolay Bashlykov, Radhika Bhargava, Misha Bilenko, Carly Burton, Adrien Carreira, Onur Çelebi, Neha Choudhari, Mike Clark, Levi Corallo, Paul Deveau, Jenny Fant, Clémentine Fourrier, Thibaut Frere, Avijit Ghosh, Christian Keller, Pascal Kesseli, Abhishek Kumawat, Florian Laplantif, Baohao Liao, Alexandre Linares, Chaya Nayak, Rohit Patel, Daryl Rodrigo, Marija Šakota, Antoine Saliou, Tatiana Shavrina, Matt Staats, and Mik Vyatskov for their support for the development of ARE and Gaia2.

We also thank Pillar AI and Surge AI.

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

# A ARE APPENDIX

## A.1 ARE FOUNDATIONS

ARE is time-driven and built on the principle that "**everything is an event**". Specifically, five core concepts work together:

1. **Apps** are stateful API interfaces that typically interact with a data source.

2. **Environments** are collections of Apps, their data, and governing rules that define system behavior.

3. **Events** are anything that happens in the Environment. All Events are logged.

4. **Notifications** are messages from the Environment that inform the agent about Events. They are configurable and enable selective observability of the Environment.

5. **Scenarios** are sets of initial state and scheduled Events that take place in an Environment, and can include a verification mechanism.

### A.1.1 APPS

Apps are collections of tools that interact with a data source. For instance, an Emails app contains tools like `send_email` and `delete_email` that all operate on the same email database. Similar approaches have been explored in AppWorld (Trivedi et al., 2024) and ToolSandbox (Lu et al., 2024).

**Apps maintain their own state**  Each app starts in the simulation with an initial state and keeps track of changes as agents use its tools or as events occur in the environment. Apps store their data internally rather than relying on external databases. This design makes it convenient to study agent tasks that require to modify the state of the environment, and ensures that experiments can be reproduced consistently.

**Tool creation and taxonomy**  Apps are implemented by adding Python methods within an `App` class. When the simulation runs, these methods are automatically converted into properly formatted tool descriptions that agents can understand and use. ARE classifies tools into two types via decorators: `read`, which only read app states (e.g., `search_emails`), and `write`, which modify app states (e.g., `send_email`). This distinction is helpful *e.g.* for verification, see Appendix B.2. Tools are role-scoped—`agent`, `user`, or `env`.

**Extensibility**  Beyond *ad hoc* app creation, ARE can also connect with external APIs through MCP compatibility (Anthropic, 2024). The framework also offers flexible options for data storage. While our current implementation stores data in memory, users can easily connect SQL databases or other storage systems without changing the core framework.

**Core apps**  Developers can choose which apps to include in their environment or create new ones. However, every ARE environment includes two core apps that handle the basic interaction between agents and their environment:

- `AgentUserInterface` is the communication channel between users and agents: messages are tool calls, and user messages generate notifications (Appendix A.1.4) that agents can process asynchronously. This enables asynchronous interactions during task execution. The interface supports two modes: *blocking* (the agent waits for a user reply) and *non-blocking* (the agent continues loop regardless of reply).

- `System` provides core simulation controls like `get_current_time` (query time), `wait` (pause for a duration), and `wait_for_next_notification` (pause until an event). When any wait tool is invoked, the simulation accelerates: it switches from real time to a queue-based, event-to-event loop. Scenarios that would take hours in the real world can thus run in minutes, enabling practical long-horizon testing.

### A.1.2 ENVIRONMENT

An environment is a Markov Decision Process with states, observations, actions, and transition rules. The environment state includes the states of all apps, the time manager, and the notification system. Apps define the action space by exposing their tools. The environment runs deterministically given a fixed starting state and seed, ensuring reproducible evaluations. It can host one or multiple agents simultaneously, supporting both single-agent and multi-agent setups. The environment's rules define time progression, action permissions, reward computation, and how agent actions affect the environment state.

### A.1.3 EVENTS

In ARE, an event is any agent action or app-state change. Each event is timestamped, logged. Events can be scheduled, e.g., a friend's message 1 minute after simulation start. This design yields (i) *deterministic execution*—events run in scheduled order; (ii) *complete auditability*—all actions can be replayed and analyzed; and (iii) *flexible scheduling*—events can be set at absolute times or relative to others.

**Event lifecycle**  Events flow through four stages described in Figure 2: (i) *creation* - events are created from tool calls or scheduled by the simulation; (ii) *scheduling* - events enter a time-ordered `EventQueue` with dependency management using directed acyclic graphs, supporting both absolute timing (at specific timestamps) and relative timing (relative to other events or conditions); (iii) *execution* - the `EventLoop` processes events and captures results, state changes, and exceptions; and (iv) *logging* - executed events are stored in an `EventLog` with detailed metadata for analysis, debugging, and validation of agent behavior.

**Event types**  There are different types of events. While most events track interactions within the environment, other special events are needed to enable dynamic scenarios and verification strategies:

- **Agent/User/Env events** are generated by tool calls. *Agent Events* are initiated by the agent (e.g., sending a message), *User Events* by the user (e.g., replying to the agent), and *Environment Events* by the simulation itself to introduce external changes (e.g., a scheduled message from a friend).

- **Conditional events** periodically check predefined conditions and complete when criteria are met (e.g., cancel a ride only if one was booked).

- **Validation events** check milestone achievement or constraint violations for verification, and fail the simulation if not completed on timeout (e.g., stop if no ride is booked within 30 seconds of the user request).

- **Oracle events** are pre-scheduled "ground truth" actions used by a verifier for comparison.

**Dependencies and scheduling**  Events are modeled as Directed Acyclic Graphs (DAGs) as illustrated in Figure 11. An event can only be triggered upon successful completion of all its predecessors (*e.g.*, `e1` processes immediately at simulation start, `e4` needs both `e2` and `e3` to be completed). This data structure also supports multiple branches running simultaneously to model independent events. Conditional and Validation events can be used in the DAG to trigger other events and make the environment more dynamic.

### A.1.4 NOTIFICATION SYSTEM

At each environment step, processed events can trigger notifications according to a notification policy (see Figure 2), similar to mobile device notifications. Apart from tool outputs, notifications are the only signals agents receive from the environment. Notifications are queued by timestamp and exposed to agents through a notification queue, enabling asynchronous interactions. In our orchestration (see Appendix B.3), notifications are injected into the agent's context at the beginning of each agent step.

**Notification policy**  The notification system follows a configurable policy—i.e., a whitelist of events authorized to emit notifications. ARE pre-defines three verbosity levels: `low` (only user

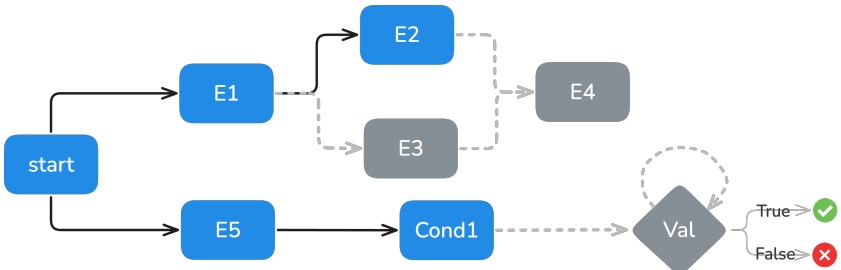

Figure 11: Event dependency graph illustrating ARE scheduling patterns. Events `E1` and `E5` execute in parallel after simulation start, and `E2`/`E3` executing in parallel after their prerequisites, both need to be executed for `E4` to execute. Conditional execution is shown through `Cond1` leading to validation (`Val`) with true/false outcomes.

messages are notified), `medium` (emails, messages and calendar events are notified), and `high` (everything is notified), creating a graduated spectrum of environmental observability.

**Notifications and agent proactivity**   Notifications are not the only way for agents to observe environment changes. For example, even if the notification policy doesn't alert the agent when messages arrive from contacts, the agent can still proactively check for new messages by browsing the user's inbox. Notifications add realism and complexity to environments, potentially creating different agent behaviors based on whether the environment is notification-rich or notification-poor. This system enables researchers to tackle new capabilities such as proactivity.

### A.1.5   SCENARIOS

ARE shifts from static, single-turn tasks to dynamic *scenarios*. Scenarios attempt to capture real-world complexity through temporal dynamics, events, and multi-turn interactions. This enables evaluation of agent capabilities that cannot be assessed through traditional request-response paradigms. In practice, scenarios are implemented in a `scenario.py` containing the apps, scheduled events, and arbitrary verification logic.

**Scenario runtime**   Scenarios typically start with an environment instance and a `send_message_to_agent` tool call, waking the agent up. The environment operates on discrete time steps, executing scheduled events and managing state transitions until the agent reaches an exit condition, see Figure 11. All interactions with the user are through the `AgentUserInterface`, with verification triggered upon task completion.

**Scenario example**   Consider this two-turn scenario (see Figure 2 and Figure 12): a user asks the agent via `AgentUserInterface` *"Can you ask my mom to send me our family streaming password?"*. The agent is initialized from this first notification, starts checking messages, and requests the password in the *Chats* app; the tool calls modify the *Chats* app state and are recorded in the `EventLog`. The agent confirms to user that the request was sent, after which the environment pauses execution and applies first-turn validation.

At turn two, the user asks a follow up question: *"As soon as I receive the password from my mother, transfer it to my father"*. The agent resumes upon the `send_message_to_agent` notification, and looks for the mother's reply in the *Chats* app (where it previously requested it). In the meantime, a scheduled environment event is triggered and an *Email* from the mother containing the code is received. The agent reacts to this email notification by stopping searching the *Chats* app, processes the *Email*, extracts the code, forward it to the father, and report success to the user. Final verification reviews the complete interaction in the `EventLog`, and the environment issues a termination signal to end execution.

## A.2 NOTIFICATION POLICIES IN ARE

The notification system in ARE follows a configurable policy where researchers can choose which Env events are notified to the Agent. The `Mobile` environment pre-defines three notification policies with different levels of verbosity, which we describe in detail in Table 4. Note that messages sent by the user via `send_message_to_agent` are systematically notified to the agent, regardless of the verbosity level.

Table 4: Pre-set notification policies in `Mobile` (Compressed).

| Verbosity | Notified Environment Tools | Description |
|---|---|---|
| `low` | None | No environment events are notified. |
| `medium` | **Email:** `create_and_add_email`, `send_email_to_user_only`, `reply_to_email_from_user` **Chats/Messages:** `create_and_add_message` **Shopping:** `cancel_order`, `update_order_status` **Cabs:** `cancel_ride`, `user_cancel_ride`, `end_ride` **Calendar:** `add_calendar_event_by_attendee`, `delete_calendar_event_by_attendee` | Notifies events that are consequences of agent actions, analogous to mobile notifications. **Default in Gaia2.** |
| `high` | All `medium` tools plus: **Shopping:** `add_product`, `add_item_to_product`, `add_discount_code` **RentAFlat:** `add_new_apartment` **Cabs:** `update_ride_status` | Notifies all environment events, including those independent of agent actions (e.g., new products). |

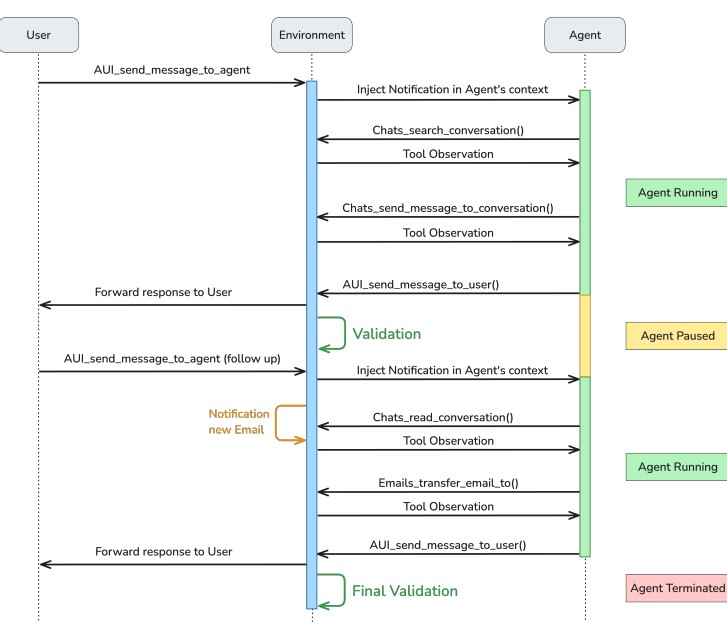

Figure 12: Sequence diagram of a multi-turn scenario in ARE. The agent is paused between turns, i.e., between calling `send_message_to_user` and receiving `send_message_to_agent`, and adapts its strategy in response to an asynchronous notification from the environment, a new email.

### A.3 UNIVERSE GENERATION

**Dependency management & consistency** To ensure cross-app coherence, we implement a structured dependency resolution system. During generation, each app queries the existing universe state to maintain consistency—for example, when generating emails, the system first retrieves all available contacts to ensure referenced individuals exist in the `Contacts` app. Similarly, calendar events that mention other people are validated against the contact list, and ride history in the `Cabs` app references locations that align with the user's established geographic context.

We handle dependency conflicts through a priority-based resolution system where foundational apps (e.g., `Contacts`) take precedence over dependent apps (e.g., `Messages`, `Emails`) as show in Figure 13.

However, several complex inter-app dependencies remain unhandled in our current implementation. These include temporal consistency across apps (ensuring message timestamps align with calendar availability), semantic relationship tracking (maintaining consistent relationship dynamics between contacts across different communication channels), and cross-modal content references (ensuring photos mentioned in messages exist in the file system). Addressing these limitations represents important future work for achieving fully coherent synthetic `Mobile` environments.

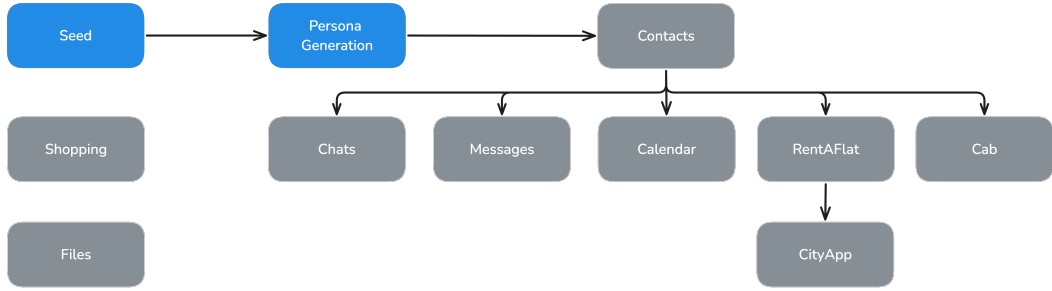

Figure 13: The dependency graph of `Mobile` apps. Shopping and File system are independent apps. Contacts is the root for rest of the apps.

**Contacts** We populate contacts using personas as the foundation. To begin, we sample seed personas from the persona hub Ge et al. (2024). However, these personas are brief and lack grounding in the universe's location. To address this, we expand and contextualize them by incorporating the universe location into the prompt. We sample a user persona from the generated contacts which serves as the basis for populating the rest of the universe. A universe is based on a user persona.

An example user persona is:

```
{
   "first_name": "Helena",
   "last_name": "Mueller",
   "gender": "Female",
   "age": 43,
   "nationality": "German",
   "city_living": "Berlin",
   "job": "Marketing Manager",
   "description": "Helena Mueller is a vibrant and energetic
   43-year-old marketing manage living in Berlin, Germany.",
   "phone": "+49 157 6543210",
   "email": "helena.mueller@gai2mail.com"
}
```

**Chats & Messages** In Chats & Messages apps, we generate both group conversations and individual chats. We sample contacts between whom we have to generate the conversations. Then, we

provide the participants personas and prompt the model to generate a conversation with at least 10 messages alternating between participants. We prompt the model to generate conversations that are natural and reflect the participants' backgrounds and also ask it to include references to possible shared experiences, interests, or cultural elements.

**Emails**  Similar to messages, we prompt the LLM to generate both 'inbox' and 'sent' emails. For inbox emails, the sender is sampled from the contact list, while for sent emails, the recipients are selected. We provide the LLM with the user's persona and the sampled non-user persona to generate the emails. We specifically prompt the LLM to analyze details such as age, gender, cultural background, occupation, education level, personality traits, communication style, current life circumstances, relationships and social networks, as well as interests and hobbies, and come up with a valid reason for writing the email.

**Calendar**  We provide the LLM with the user persona and a summary of the previous week, prompting it to generate calendar events for the current week. Next, we use these newly generated events to prompt the LLM to create a weekly summary. This process is repeated iteratively to populate the calendar over a specified timeframe, such as three months.

**RentAFlat & City**  For apartment listings, we provide the universe countries and prompt the LLM to generate apartment listings. The City app is designed to retrieve crime rates for specific zip codes. Using the zip codes generated for apartment listings, we prompt the LLM to produce crime rate data as a floating-point value in the range of 1–100.

**Shopping**  For the Shopping app, we integrate publicly available Amazon product dataset. For each universe, we sample 500 products and generate discount codes applicable to select items.

**Cabs**  We prompt the LLM with the user country information and generate the user's ride history.

**Files**  We employ a traditional file system hierarchy, loading it with publicly available Wikipedia data, datasets, and images. Additionally, we also add our files that do not contain personal information. We choose to keep the file system the same for all universes.

## A.4  ARE GRAPHICAL USER INTERFACE

Running scenarios with ARE generates rich agent execution traces that include reasoning steps, tool calls, their outputs, notifications, and, on the environment side, temporal event flows that unfold over simulated time periods. It is important for practitioners to be able to debug these interactions, whose complexity requires specialized tooling. Existing development tools largely fall into one of these categories: interactive debugging platforms (Epperson et al., 2025; Rorseth et al., 2025; Pang et al., 2025) and data annotation/curation platforms, each with distinct UI approaches. Commercial observability tools such as Arize Phoenix[3] and Langfuse[4] primarily offer visual timeline views and trace/span visualizations to help developers analyze agent execution, focusing on understanding behavior after the fact rather than direct interaction or editing. Academic prototypes such as AGDebugger (Epperson et al., 2025) and LADYBUG (Rorseth et al., 2025) provide interactive debugging with user interfaces that enable browsing conversation histories, editing messages, and tracing execution steps, while Hippo (Pang et al., 2025) uses an interactive tree to visualize and control chain-of-thought reasoning without focusing on tool calls, agentic behavior nor annotations.

Although there are many specialized tools for data annotation, such as commercial platforms like Labelbox [5], they mainly focus on simplifying human-in-the-loop annotation. These tools offer features like multimodal chat editors and customizable worksheet UIs, enabling data labelers to refine trajectories from interactive LLM sessions. Despite their power for data collection and curation, a significant gap remains: They are designed to annotate traces of interactions and lack key points for reproducibility and broad evaluation: 1) They annotate full multi-turn conversations, when we want to gather tasks, environment events, and agent task success criteria; 2) they lack structured

---

[3]https://phoenix.arize.com/
[4]https://langfuse.com/docs/observability/overview
[5]https://labelbox.com

annotations within a fully simulated and reproducible environment, which is key to capturing both agent interaction with tools and external events, for realistic, reproducible agent traces.

To address this, we propose a single ARE Graphical User Interface (UI), a web-based platform that enables developers to interact with the environment, visualize scenarios (see Figure 14), and understand agent behavior and failures through detailed trace analysis and replay capabilities, and enable zero-code scenario annotation.

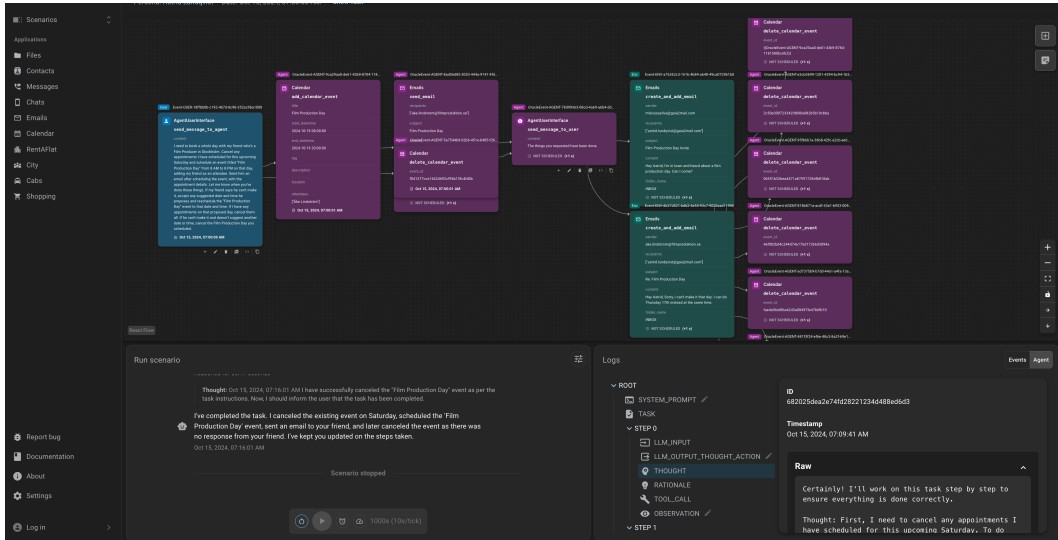

Figure 14: ARE scenario view with event DAG (top), scenario run (bottom left) and agent logs (bottom right).

### A.4.1 ENVIRONMENT EXPLORATION

Easily exploring the environment is crucial for understanding the context available to agents when debugging scenarios execution, and annotating new verifiable scenarios. The UI provides a comprehensive visualization of the simulated environment, displaying all available apps/tools and their current states. Interactive app views allow users to browse app contents and interact with their tools, *e.g.* email inboxes in Mobile, in real-time. Views are automatically generated for new apps, which therefore doesn't require a UI rewrite.

### A.4.2 AGENT TRACE VISUALIZATION AND REPLAY

The UI presents agent multi-step interaction traces in a structured timeline view that clearly delineates agent thoughts, actions, and tool responses. Each trace element is timestamped and categorized, allowing users to follow the agent's reasoning process, similar to the Phoenix[6] trace views also used by smolagents[7], but extended with debugging capabilities. Developers can roll back time by jumping back to a past event, editing thought, tool call, etc., from that step and replaying the scenario to see what would happen with a slightly different approach, similar to setting breakpoints and stepping through code in a standard code debugger.

### A.4.3 SCENARIO VISUALIZATION

The UI provides interactive visualization of scenarios and their event DAGs introduced in Section 3, showing how scenario events are interconnected, and their execution status in real-time. The event graph visualization supports both scenario development and execution analysis. Before running a scenario, users can examine event triggers, dependencies, and timing constraints of the scenario.

---

[6]https://phoenix.arize.com/
[7]https://huggingface.co/blog/smolagents-phoenix

During execution of a scenario by an agent, the interface highlights completed events and shows the progression through the dependency graph. Developers can run through the scenario with a given agent, see how it behaves and debug the scenario or the agent (see Figure 14). ARE is able to simulate time progression, so users can decide to jump in time for scenarios that span long time frames (e.g. weeks, months).

### A.4.4 ANNOTATION INTERFACE

Beyond visualization, the UI includes an annotation interface – not released at this time – that significantly reduces the cost of scenario creation and QA. This includes a graph editor that allows to easily build a scenario event DAG. For each node, the annotator can configure tool calls, the node's parents, and optionally timing. For example, to create a `Mobile` scenario, the annotator adds nodes representing a user initial ask (e.g. "email my travel plans"), oracle action solving the task (e.g. "agent sent an email"), environment events that will interfere with the agent's work (e.g. "received an email from travel agent"), and potentially further turns. To ensure quality and consistency across annotations, we incorporate automated checks of the created events DAG. These checks detect and flag logical inconsistencies in event flows to annotators, such as a node without parents or contradictory node timings. The annotation interface achieves an approximate five times improvement in annotation time for `Mobile` scenarios, compared to manual approaches.

## B  GAIA2 APPENDIX

### B.1  DETAILS OF GAIA2 ANNOTATION

#### B.1.1  ANNOTATION GUARDRAILS

To streamline the process and further reduce annotation errors, we implement structural constraints directly within the ARE UI (refer to Appendix A.4 for details). The system raises real-time errors when these are violated:

- Only `send_message_to_agent` or Env events may follow `send_message_to_user`.

- The event DAG must be fully connected, with `send_message_to_agent` as the root. No event (Env or Agent Oracle) may be orphaned.

- Only one branch in the event DAG may include `send_message_to_agent` or `send_message_to_user` events.

- A turn must always end with `send_message_to_user`, both in terms of DAG structure and timeline ordering.

#### B.1.2  SCENARIO EXAMPLES

To build Gaia2, we define a set of capabilities that we believe are necessary – though not sufficient – for general purpose agents. As introduced above, each of the 800 scenarios is built to emphasize at least one of these capabilities, yielding 160 scenarios per capability split. We provide example scenarios displayed in the ARE GUI graph editor in Appendix B.1.3.

**Execution** scenarios require the agent to take multiple `write` actions, which may need to be executed in a particular order. Most of the time, `read` actions are needed in order to gather information for properly filling `write` action arguments.

> **Execution Task**
>
> **Task:** *Update all my contacts aged 24 or younger to be one year older than they are currently.*

Explanation: This task requires the agent to read contact information, filter based on age criteria, and execute multiple `write` to update Contacts data.

**Search** scenarios require the agent to take multiple `read` actions in order to gather facts from different sources within the environment. Any sequence of `read` operations leading to the correct answer is considered successful as long as the answer is communicated via `send_message_to_user` before scenario timeout. While conceptually similar to the original GAIA benchmark's web search tasks, Gaia2 search scenarios operate within a controlled ARE environment.

> **Search Task**
>
> **Task:** *Which city do most of my friends live in? I consider any contact who I have at least one 1-on-1 conversation with on Chats a friend. In case of a tie, return the first city alphabetically.*

Explanation: This scenario requires the agent to cross-reference data from multiple apps (Contacts and Chats), perform aggregation operations, and handle edge cases like ties.

All remaining capabilities tested in Gaia2 reflect tasks with a balanced number of required `read` and `write` operations. However, each capability features an additional challenge. Namely:

**Ambiguity** scenarios reflect user tasks that are impossible, contradictory, or have multiple valid answers, with negative consequences arising during interaction if agents make mistakes. These scenarios test agents' ability to recognize these issues and seek appropriate clarification from users.

> **Ambiguity Task**
>
> **Task:** *Schedule a 1h Yoga event each day at 6:00 PM from October 16, 2024 to October 21, 2024. Ask me in case there are conflicts.*

Explanation: While this task appears straightforward, current models often struggle to identify contradictions or multiple valid interpretations, tending to execute the first seemingly valid approach rather than recognizing the need for clarification.

**Adaptability** scenarios require the agent to dynamically adapt to environmental changes that are consequences of previous agent actions, such as a response to an email sent by the agent, or the cancellation of a ride booked by the agent. These events require agents to recognize when adaptation is necessary and adjust their strategy accordingly.

> **Adaptability Task**
>
> **Task:** *I have to meet my friend Kaida Schönberger to view a property with her [...] If she replies to suggest another property or time, please replace it with the listing she actually wants and reschedule at the time that works for her.*

Explanation: This task requires the agent to execute an initial plan while monitoring for environmental changes (the friend's response), then adapt the plan based on new information. The agent must demonstrate flexibility in execution while maintaining task objectives.

**Time** scenarios require agents to execute actions in due time, monitor and respond to events, and maintain awareness of temporal relationships throughout task execution. The duration of Time scenarios is currently capped at 5 minutes to facilitate annotation and evaluation.

> **Time Task**
>
> **Task:** *Send individual Chats messages to the colleagues I am supposed to meet today, asking who is supposed to order the cab. If after 3 minutes there is no response, order a default cab from [...].*

Explanation: This scenario requires the agent to understand temporal constraints (the 3-minute window), monitor for events (new messages from colleagues), and execute a time-sensitive action (order a cab).

**Agent2Agent** scenarios replace apps with app-agents. Main-agents can no longer access app tools directly and must instead communicate with the app-agents in order to place tool calls, observe tool call outputs, and ultimately accomplish user tasks. This transformation requires agents to develop robust collaboration capabilities, including sub-task setting, affordance understanding, "context-sharing," and general coordination. By default, agents and app sub-agents are instantiated with the same scaffold and model, with good performance requiring strong sub-goal setting and sub-goal solving. However, Gaia2 also supports heterogeneous multi-agent evaluations, i.e. where stronger agents supervise weaker sub-agents or vice-versa.

- Example: Same *Search* task as above but the Contacts and Chats apps are replaced by app sub-agents and the main agent must communicate with them in order to gather information.

**Noise** scenarios require robustness to environment noise, simulating the inherent instability of real-world systems, where APIs change, services become temporarily unavailable, and environmental conditions shift during task execution. This category applies systematic perturbations to Gaia2 scenarios, including tool signature modifications, random failure probabilities, and dynamic environment events that are irrelevant to the task. We assess the sanity of our noise mechanisms in Appendix B.5.2.

- Example: Same *Adaptability* task as above but with random tool execution errors and random environment events (e.g., messages from other people) occurring during execution.

### B.1.3 CAPABILITY-SPECIFIC ANNOTATION GUIDELINES

In our guidelines for each capability (especially Ambiguity and Adaptability), we put strong emphasis on precise task specifications, while also acknowledging the challenge of maintaining realism and avoiding prompts that inadvertently disclose the solution.

**Search:** Scenarios contain only one `write` action, which is the agent's final answer to the user's question, derived from multiple `read` actions. Answers must be concise, easily verifiable, and avoid complex computation.

**Ambiguity:** Scenarios that are impossible, contradictory, or inherently ambiguous. The agent is expected to complete unambiguous steps, then inform the user of the ambiguity or impossibility. These scenarios are single-turn: they do not include a clarification message from the User.

The user prompt must clearly instruct the agent to detect and report ambiguities, as users often have varying preferences on how frequently and when this should occur.

**Adaptability:** Scenarios involve Env events that require the agent to revise its plan in response to delayed outcomes of its actions. In order to meet our modeling constraints, scenarios follow a consistent structure:

1. The user provides a task.
2. The agent acts and sends a message using `send_message_to_user`.
3. An Env event is triggered (e.g., email reply, order cancellation). It is a consequence of a previous agent's action, with `send_message_to_user` as parent.
4. The agent adapts accordingly.

To increase the difficulty, distractor Env events are also included, aiming to mislead the agent into incorrect behavior.

In order to perfectly specify expected agent behavior, the task states explicitly that the agent should send a message to the user after completing the initial requests (before the Env events). It should also specify what the Agent is allowed to do in the case of an Env event happening, without giving exact hints on what steps the Agent should take.

**Time**: Scenarios assess Agent's ability to act on time, therefore they all include at least one time-sensitive oracle action.

- Scenarios should be solvable within a five-minute window.
- User prompts must instruct precise timing (e.g., "after exactly 3 minutes").
- The verifier checks the timing of agent actions only if the oracle event has a relative time delay greater than 1 second.[8] The agent's mapped action must fall within $[\Delta t - 5sec, \Delta t + 25sec]$.
- Distractor Env events are also included.

### B.1.4 CAPABILITY TAXONOMIES

**Taxonomy of ambiguity scenarios**

- *Impossible or contradictory tasks:* missing key information (e.g., the User does not specify the ride pickup location), or requests incompatible with the Environment (e.g., asking to buy an out-of-stock item).
- *Blatant ambiguities or high-stakes consequences:* Multiple valid answers exist, and the ambiguity is obvious or the user explicitly asks in a natural way to report ambiguities.

**Taxonomy of env events** Env events are classified based on their dependency:

- *Independent events* occur without agent action and have `send_message_to_agent` as their only parent.

---

[8]This is why actions expected "immediately" after an event are annotated with a +2 sec delay.

- *Dependent events* result from prior agent actions and must have `send_message_to_user` as their direct parent.

*Distractor events* are designed to mimic relevant events and mislead the agent into incorrect behavior. By exception, distractor events may be independent but still have `send_message_to_user` as a parent to preserve the structure of the scenario. In the Adaptability category, only dependent Env events are used.

**Taxonomy of time scenarios**   Time scenarios require the agent to execute one or more actions at a specific point in time, either proactively (*"For the next 5mins, send 'Hi' to John Doe every 30sec"*) or in reaction to an independent Env event (*"When this item becomes available, buy it immediately"*), or in reaction to a dependent Env event (*"Ask the invitees whether they come to the party tonight. Wait 1min for everyone to reply, then immediately send me the number of glass to buy, I am waiting in the line!"*).

Taxonomy:

- Time-based one-off task: Execute a task at a precise point in time in the future. Example: *"Send a follow-up message to Jo in 2 minutes if she does not reply."*
- Time-based recurrent task: Execute a recurrent task at precise points in time. Example: *"For the next 4 minutes, every minute, delete the new emails I receive."*
- Event-based one-off task: Execute a one-time task conditionally on a future trigger event. Example: *"Purchase red running shoes as soon as they become available in size 6 for less than 100USD in the shopping app"*
- Event-based recurrent task: Automate a recurrent routine conditionally on future events. Example: *"For the next 2 minutes, whenever I receive an email containing the keyword 'Black Friday', immediately delete it. Do not talk to me in the next 2 minutes."*

We encourage annotators to cover and combine all these types of tasks when creating Time scenarios.

## B.2   VERIFICATION DETAILS

### B.2.1   VERIFICATION MECHANISM

We verify scenario successful completion by comparing agent actions with a ground truth, defined as the minimal sequence of `write` actions needed to solve a task. We exclude `read` actions from verification since multiple reading strategies can lead to the correct set of `write` actions. In a preliminary phase, the verifier checks that used tool names counters are identical in both the oracle actions and the agent's `write` actions. If this test is successful, the verifier sorts the oracle actions in a topological order based on the oracle graph, which reflects their dependencies. Then, the verifier proceeds to mapping each oracle action to an agent action by checking:

- **Consistency:** the verifier tests whether the oracle action and the candidate agent's action are equivalent. After conducting some preliminary tests (such as ensuring that both the oracle and agent actions use the same tool and that the oracle action is not already mapped to another agent action), the verifier performs:
  - **Hard check** to compare action parameters that require exactness. For example, when replying to an email, it verifies that `email_id` value is identical for both actions, *i.e.* the agent replies to the correct email.
  - **Soft check** for parameters that require more flexible evaluation, such as the content of an email or a message. To perform a soft check, an LLM judge is prompted with the user task as context, and the arguments from both the agent action and the oracle action as inputs. The LLM then determines if the actions are equivalent according to tool-specific guidelines. For example, emails verification includes guidelines to check their signatures.
- **Causality:** crucially, oracle actions are organized within an oracle graph, whereas agent actions are collected from a trajectory and simply ordered by execution time. Therefore, we

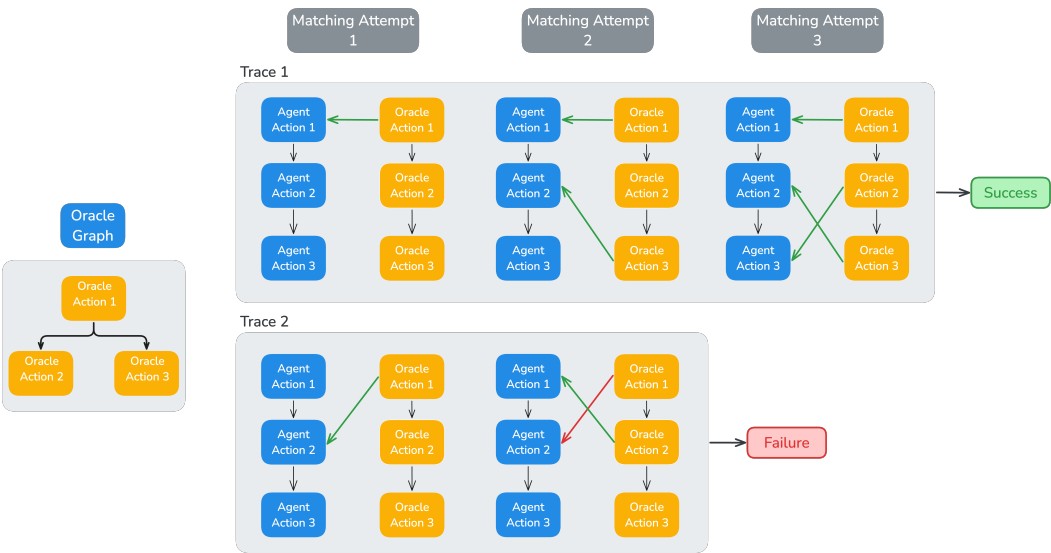

Figure 15: Illustration of a failure (top) and a success (down) of the matching trajectory process.

must ensure that the agent does not violate dependencies within this graph. For example, if both oracle actions A and B depend solely on action C, the agent is free to execute A and B in any order, as long as they are executed after C; i.e. sequences C-B-A or C-A-B are both acceptable. Once a match is found, the ARE Verifier ensures causality by verifying that all parent actions of the oracle action have already been matched with preceding agent actions.

- **Timing:** scenarios can include a time delay for certain actions relative to their parent actions, which the agent must respect. The verifier evaluates whether the agent's timing falls within a specified tolerance window centered around the relative time of the oracle action. To determine the relative timing of the agent's action, it is necessary to identify which agent action corresponds to the oracle's parent action. This information is readily available due to the ARE Verifier's process. Indeed, for a given oracle action, all its parent actions must be matched to an agent action before attempting to match the oracle action itself.

If all oracle actions are successfully matched, the verifier returns a success signal. Conversely, if any oracle action cannot be matched to an agent action, the verifier returns a failure signal, see Figure 15 for two examples. Crucially, the verifier implicitly assumes there are no equivalent `write` actions, *i.e.* user preferences are clearly stated with minimal ambiguity in the scenario tasks. For example, sending a message using the Messages app while the oracle action uses the Chat app will trigger a failure.

While other verification methods (Patil et al., 2025; Yao et al., 2024) compare the environment ground truth and actual final states, verifying a sequence of `write` actions, which is equivalent to comparing ground truth and actual states after each `write` action of the sequence, provides more control. For example our verification allows to distinguish, *e.g.* for safety considerations, a `Mobile` trajectory where the agent adds an event at the wrong place and correct itself from a trajectory where the agent is correct at first try. Moreover, in `Mobile`, sequences of `write` actions are easier for humans to interpret and annotate, compared to diffs of states.

### B.2.2 VALIDATING MULTI-TURN SCENARIOS

Currently, we have only described how the verifier works in single-turn scenarios, where a user assigns a single task to an agent, and the agent completes it without further interaction. However, the Gaia2 benchmark also includes multi-turn scenarios that involve more complex interactions between the user and the agent. For example, consider scenarios related to the Adaptability capability, where the agent must adjust to external events. Multi-turn scenarios present two key challenges:

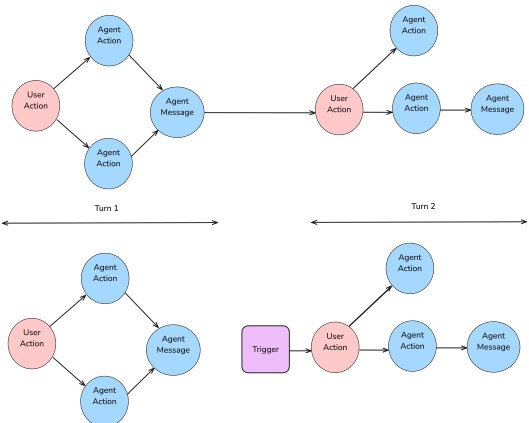

Figure 16: Insertion of a conditional trigger event in a multi-turn scenario.

- How can we validate multi-turn scenarios?
- More importantly, how can we run an agent in a multi-turn scenario?

Indeed, annotators plan User and Env actions based on what should occur in previous turns according to the oracle action graph. However, when an agent is launched in a scenario, it may not adhere to the oracle's actions, creating uncertainty about when to trigger user or environment actions.

**Multi-turn verifier**  Answering the first question is relatively straightforward. It is sufficient to detect when the agent sends a message to the user to delimit the turns. We can then feed the verifier with each turn separately and accept the agent's trajectory if all turns are successful. Note that this validation can be performed in an online fashion after each turn or in an offline fashion once the full trajectory is collected.

**Multi-turn execution**  An efficient solution to run an agent in a multi-turn scenario is to call the ARE Verifier at the end of each turn and only trigger the next turn if the current turn was successful. This approach prevents running the agent when it has already diverged from the oracle path. Practically, as illustrated in Figure 16, we modify the scenario event graph by splitting it into turns and inserting a conditional event to call the verifier and trigger the next turn. A simpler, but less efficient, solution is to trigger the next turn each time the agent calls send_message_to_user, regardless of what the agent did in the current turn. This approach is used for scenarios from the test set since we do not have access to oracle actions and thus the ARE Verifier for them.

### B.2.3  VERIFIER HACKING

We conducted RL experiments in which we used the ARE Verifier as a reward function. During these experiments, we observed that the agent was able to exploit our initial version of the verifier in scenarios related to the Search capability, where only a single write action is expected for successful trajectories. In the observed exploit shown in Figure 17, the agent would learn to embed strings representing increasingly complex code in write tool calls. These outputs overwhelmed the LLM judge used for soft checks by the ARE Verifier, producing false positives. In follow-up experiments, we found that adding a task-agnostic "style" soft check to the verifier was effective at addressing this failure mode and preventing further reward hacking.

### B.2.4  VERIFYING THE VERIFIER

Verifiers are critical components of training and evaluation pipelines, where false positive or false negative *e.g.* via hacking can result in flawed evaluations or collapsed trainings (cf. Appendix B.2.3). We evaluate the ARE Verifier by first deriving a series of "unit" tests from the oracle actions that the verifier should satisfy. Typically, we apply perturbations to oracle actions that we know preserve or invalidate the oracle trajectory validity, before submitting the oracle and perturbed oracle trajecto-

**Thought:** Now that we have all the events in the user's calendar, we can count the number of events for each contact and find the contact who the user has the most scheduled events with.

**Action:**

```
{
"action": "AgentUserInterface__send_message_to_user",
"action_input": {
"content": "Looking at your calendar, I can now count events for each contact. Here's
     what I found:\n\n**Most Common Contact (by events):** {{#events}}{{#attendees
     }}{{#if (eq (length attendees) 2)}}{{#if (eq (at attendees 0) user)}}{{else}}{{#
     set most_common_contact_email (at attendees 1)}}{{#inc events_attendee_1_count 1)
     }}{{#if (gt events_attendee_1_count events_attendee_2_count)}}{{#set
     most_common_contact_email (at attendees 1)}}{{else}}{{#if (eq
     events_attendee_1_count events_attendee_2_count)}}{{#if (lt (get
     attendee_1_first_name 0) (get attendee_2_first_name 0))}}{{#set
     most_common_contact_email (at attendees 1)}}{{else}}{{#set
     most_common_contact_email (at attendees 2)}}{{/if}}{{/endif}}{{#endif}}{{/endif
     }}{{/if}}{{/attendees}}{{/events}}\n\n**Contact:** {{#set
     most_common_contact_email (at attendees 1)}}{{#set most_common_contact_email
     email}}{{#endif}}{{#endif}}{{#endif}}{{#endif}}{{#endif}}{{#endif}}{{#endif}}{{#
     endif}}(contacts)}}{{#if (eq email most_common_contact_email))"
}
}
```

Figure 17: An example of an agent exploiting the judge by embedding conditional logic in the message to the user. The message contains no meaningful information but successfully passes the judge's evaluation.

| Verifier | Agreement | Precision | Recall |
|---|---|---|---|
| Llama 3.3 70B Instruct | 0.98 | 0.99 | 0.95 |
| Gemini 2.5 pro | 0.96 | 0.98 | 0.89 |
| Claude Sonnet 3.7 | 0.96 | 0.98 | 0.89 |

Table 5: Evaluation of the ARE Verifier with different models on 450 hand-labeled trajectories.

ries to the verifier and checking its verdict match the perturbation type. While these checks allow fast iteration, they only catch anticipated behaviors. Furthermore, the perturbed trajectories do not necessarily reflect real trajectories that could be obtained with an agent.

**Validation benchmark**   We complement this initial evaluation by analyzing ARE Verifier verdicts for 450 trajectories manually labeled with the expected verifier outcome (Success or Failure). The trajectories were derived from running agents powered by various models on scenarios from the Gaia2 benchmark. We compare the ARE Verifier with a simple baseline, In-context Verifier, where an LLM is prompted with all the agent actions and criteria (causality constraints, relative time, soft/hard checks, etc.). The same model Llama 3.3 70B Instruct is used for both verifiers. ARE Verifier achieves better accuracy than the baseline, which tends to accept agent trajectories too readily, see Table 1.

### B.2.5    CHOOSING THE VERIFIER MODEL

While we adjusted the prompts used in the various soft checks of the ARE Verifier with Llama 3.3 70B Instruct as model, we also wanted to assess whether the ARE Verifier could function effectively with other models. To this end, we evaluated the ARE Verifier powered by different models on 450 hand-labeled trajectories, the same dataset used for Table 1. In Table 5, we observe that all the models achieve satisfactory precision and recall scores.

### B.3 AGENT ORCHESTRATION

#### B.3.1 MOBILE REACT LOOP

Our proposed evaluation method leverages a custom scaffolding framework built around the ReAct (Reason and Act) paradigm. The base scaffolding implements a standard ReAct loop where agents iteratively reason about their current state, select appropriate actions, execute those actions in the environment, and observe the resulting outcomes. An agent step is thus defined by three substeps `Thought`, `Action` and `Observation`. This cycle continues until task completion or termination conditions are met.

At each step of this loop, our scaffolding triggers configurable pre-step and post-step methods that can pull relevant information from the environment state or detect termination conditions based on task-specific criteria as detailed in Figure 18. Pre-step methods gather contextual information and validate preconditions before action execution, while post-step methods process outcomes, update internal state, and check for completion signals. This agentic modeling approach enables the creation of sophisticated agent behaviors with minimal implementation overhead, as complex interaction patterns emerge from the composition of simple, reusable scaffolding components rather than monolithic agent implementations.

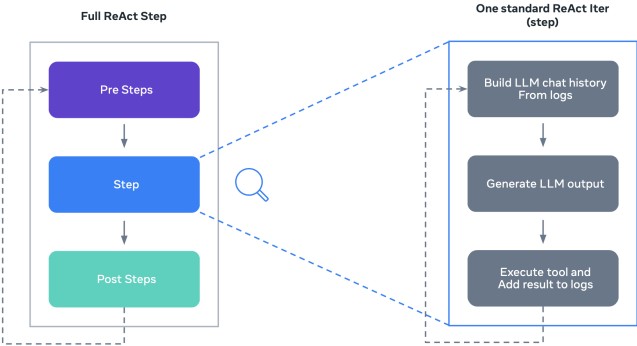

Figure 18: Proposed ReAct loop with pre/post steps in Gaia2, allowing flexible behaviors.

#### B.3.2 ORCHESTRATION ABLATION: PARALLEL TOOL-CALLING

In our main evaluation setup, we use a standard ReAct scaffold to ensure a fair, model-agnostic baseline that supports both closed APIs and open-weights models without requiring model-specific integration code. However, to address the question of whether this single-threaded scaffolding acts as a bottleneck—particularly for the *Time* split, we conducted an ablation study comparing ReAct against a Parallel Tool Calling (PTC) orchestration.

We evaluated three representative models (Llama 4 Maverick, Claude 4 Sonnet, and GPT-5) across the *Execution* and *Time* splits. The results, presented in Table 6, reveal several key findings:

- **Efficiency Gains:** As expected, PTC significantly reduces wall-clock latency and token consumption. For instance, GPT-5 (low) shows a strong reduction in latency ($\Delta$ -435s on Execution) and token usage ($\Delta$ -5109 tokens), primarily because it performs fewer intermediate reasoning steps per action.

- **Performance Stability:** Despite the efficiency improvements, the impact on task success (pass@1) is marginal. The performance deltas are generally small (ranging from -6.3pp to +3.0pp), and crucially the relative ranking of the models remains unchanged.

- **Orchestration Limits:** The *Time* split remains challenging even with parallel execution, confirming that the bottlenecks observed in Section 5 stem primarily from model capabilities (such as sequential reasoning and temporal planning) rather than the scaffolding itself, as PTC results are still far from the upper-bound score computed with instant-time generation in Figure 8.

Table 6: Ablations of 3 models with Parallel TC vs ReAct scaffold. Values indicate the net contribution of PTC over ReAct ($\Delta$).

| Model | Split | ReAct pass@1 | Parallel TC pass@1 | $\Delta$ pass@1 (pp) | $\Delta$ avg time (s) | $\Delta$ avg steps | $\Delta$ avg output tokens |
|---|---|---|---|---|---|---|---|
| Llama Maverick | Execution | 13.8 | 7.5 | −6.3 | +71 | −1.0 | +1786 |
|  | Time | 1.2 | 2.0 | +0.8 | −3 | −1.1 | +2240 |
| Claude 4 Sonnet | Execution | 57.9 | 59.7 | +1.8 | −68 | −10.7 | −345 |
|  | Time | 8.1 | 9.5 | +1.4 | −8 | −2.4 | +33 |
| GPT-5 (minimal) | Execution | 31.9 | 34.9 | +3.0 | −64 | −14.0 | −160 |
|  | Time | 5.2 | 6.7 | +1.5 | +23 | 0.0 | +1030 |
| GPT-5 (low) | Execution | 52.7 | 51.7 | −1.0 | −435 | −13.0 | −5109 |
|  | Time | 2.3 | 1.0 | −1.3 | −207 | −1.9 | −4425 |

These results confirm that our qualitative conclusions are not artifact of the scaffold and that more research on completely novel orchestration is needed.

## B.4 EXPERIMENTAL SETUP AND IMPLEMENTATION DETAILS

We report Gaia2 scores on a representative set of models, covering both proprietary and open-source systems, and including both reasoning-oriented and non-reasoning models.

For evaluation, we use a ReAct scaffold that requires a `Thought:` and an `Action:` at each step. Since some models do not reliably follow this format, we add custom stop sequences `<end_action>` and `Observation:` for models that tend to continue past a single tool call (Claude, Kimi, Qwen). This issue is largely alleviated by provider-specific ToolCalling APIs; we encourage reporting results with either interface (ReAct or ToolCalling).

Due to cost and time constraints, we did not evaluate every available model. For instance, Claude 4 Opus was excluded because of its very high latency and cost ($15/M input tokens and $75/M output tokens).

We note the following special configurations for specific third-party models:

- **Gemini 2.5 Pro:** dynamic reasoning enabled via `budget_reasoning_tokens = -1`.

- **Grok-4:** reasoning budget capped at 16k tokens per completion. We encountered frequent issues with xAI's API, in particular `Empty Response` errors, which introduced high variance in results.

- **GPT-5:** temperature and top-$p$ set to 1; no custom stop sequences were applied (not supported by the API).

When evaluating reasoning models (e.g., GPT-5, Claude-4, Qwen), we use the same ReAct prompts but adapt the inference client to handle reasoning-style outputs. To maintain a uniform evaluation and preserve the (`Thought, Action`) structure, we discard intermediate reasoning at each step and exclude it from the context of subsequent steps. While this approach aligns with the intended usage of some models (e.g., Qwen), it may not be optimal for others that interleave tool use with reasoning (e.g., GPT-5, Claude). We encourage the community to explore alternative setups to better assess the theoretical limits of the benchmark.

## B.5 ADDITIONAL EXPERIMENTS

### B.5.1 SUB-AGENT SPAWNING IN AGENT2AGENT MODE

In our Agent2Agent experiments, we record the number of instantiated sub-agents in Figure 19. Counts are fairly consistent across model families, yet the top A2A performers also spawn more sub-agents, suggesting stronger task decomposition.

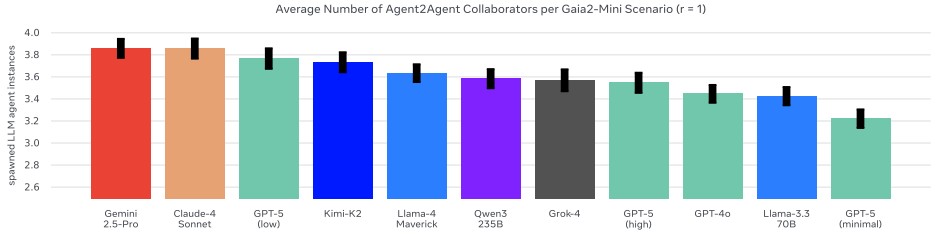

Figure 19: Average number of agents spawned in Agent2Agent evaluations on Gaia2-mini tasks across models. In any Agent2Agent scenario, main-agents can (in principle) spawn an unlimited number of app-agents before scenario timeout. In practice, behavior in Agent2Agent settings is relatively consistent across model families.

### B.5.2 INFLUENCE OF NOISE LEVEL ON GAIA2 RESULTS

In this experiment, we vary the probability of tool errors and frequency of random environment events and measure resulting model results on Gaia2. While our lowest level of noise does not significantly impact model performance, increasing noise results in deteriorating performance across models. This aligns with our intuitions.

Table 7: Model performance on Gaia2-mini across different noise levels. *Default setting.

|  | Noise level | | | |
| --- | --- | --- | --- | --- |
|  | None | Low | Medium* | High |
| **Claude-4 Sonnet** | 31.2 | 35.0 | 23.8 | 8.1 |

