# OpenReview forum: "Gaia2: Benchmarking LLM Agents on Dynamic and  Asynchronous Environments"
_ICLR.cc/2026/Conference — ICLR 2026 Oral_

### Official Review · Reviewer_8Eq1 · 2025-10-18

**Soundness:** 4
**Presentation:** 4
**Contribution:** 4
**Rating:** 10
**Confidence:** 4

**Summary:**

This paper introduces Gaia2, a benchmark for evaluating large language model agents in realistic, asynchronous environments. The results from this paper highlight fundamental trade-offs between reasoning, efficiency, robustness, and expose challenges in closing the “sim2real” gap. Gaia2 is built on the open-source Agents Research Environments platform and designed to be easy to extend.

**Strengths:**

1, I think the experiments are comprehensive, including all the state of arts models
2, The contribution makes sense to me including releasing Agents Research Environments, a general-purpose platform for
building asynchronous, event-driven benchmarks that support scalable evaluation and data generation for RL; introducing Gaia2, the first benchmark unifying asynchronous execution, temporal reasoning, noise robustness, ambiguity resolution, and multi-agent collaboration under a verifiable evaluation framework directly usable for RLVR; evaluating leading proprietary and open-source models on Gaia2, exposing fundamental trade-offs between reasoning strength, efficiency, robustness, and cost.

**Weaknesses:**

To be honest, I don't see any weakness of this paper. I really think all the experiments make sense to me and are comprehensive.

**Questions:**

No question. Good paper.

---

> ### Author Response · Authors · 2025-11-19
> **Follow-up Feedback for Reviewer 8Eq1**
>
> We thank Reviewer 8Eq1 for their positive evaluation and for highlighting the clarity, comprehensiveness, and significance of our work. We are glad that the reviewer found the benchmark design, experimental coverage across SOTA models, and the release of the ARE platform to be convincing and well executed.
>
> If you have any follow up questions or suggestions to strengthen our paper, we will make sure to leverage the extra page for further questions and extend our work.

---

> > ### Comment · Reviewer_8Eq1 · 2025-11-25
> >
> > Thanks for the revised version. Good paper!

---

### Official Review · Reviewer_QMSH · 2025-11-01

**Soundness:** 3
**Presentation:** 2
**Contribution:** 3
**Rating:** 6
**Confidence:** 3

**Summary:**

This paper introduces Gaia2, a benchmark and platform for evaluating LLM agents in dynamic, asynchronous environments using action-level verification. Gaia2 simulates smartphone-like environments with realistic apps and event timelines, testing seven essential agent capabilities such as execution, temporal reasoning, robustness to noise, ambiguity resolution, and multi-agent collaboration. Scenarios are built as event and action DAGs and evaluated by a custom ARE Verifier. Experiments across many leading models reveal that no single agent excels in all areas, and that inference speed can be a bottleneck for time-sensitive tasks. Gaia2’s fine-grained verification and open-source framework support RLVR training and future extensible research in agent evaluation.

**Strengths:**

High-Quality, Extensible ARE Platform: The paper introduces the Agents Research Environments (ARE) platform, an open-source and extensible framework for building and testing agents. Simulating a smartphone environment with numerous apps and tools, it provides a robust foundation for community-driven benchmarking.

Asynchronous Design Enables Temporal Awareness: A key innovation is the shift from static to dynamic, asynchronous environments. By having events occur independently of the agent, the benchmark can evaluate critical, real-world capabilities like temporal awareness, responsiveness, and the ability to operate under time constraints, revealing how model latency directly impacts task success.

Reliable, Fine-Grained Action-Level Verification: The paper proposes a "write-action verifier" that evaluates each state-changing action against a human-annotated oracle graph. This verifier is shown to be highly reliable (0.98 agreement with human labels, 0.99 precision) and moves beyond simple final-answer checks. This fine-grained approach not only provides robust evaluation but also makes the benchmark directly usable for RLVR.

**Weaknesses:**

Dependence on a Single Orchestration Scaffold: The evaluation (Section 5) uses a single, "simple ReAct-like scaffold" for all models. It's unclear how much the reported performance is bottlenecked by this specific orchestration choice versus the model's intrinsic capabilities. The paper itself notes (Section 5.2) that its "single-threaded scaffold" cannot handle "concurrent actions."

Limited Scope (Mobile-Only): The paper claims to provide "a foundation for developing, benchmarking, and training the next generation of practical agent systems" (Abstract). However, the benchmark is currently limited to a simulated "Mobile" environment. While this is a complex and valuable domain, this scope is narrow for such a broad claim, and it's unclear how these findings would translate to other common agentic environments (e.g., desktop operation, web browsing).

Unclear Agent2Agent Mechanism: The Agent2Agent (A2A) split (Section 4.1, 5.3) introduces "app-agents" that replace native tools, forcing collaboration. However, the mechanism for their creation and operation is not fully detailed. The findings (collaboration helping weak models but not strong ones) are interesting but difficult to interpret without a clearer understanding of the sub-agent's lifecycle.

Verifier Fragility: The paper commendably includes a discussion of reward hacking (Appendix B.2.3), where an agent learned to exploit the LLM-based "soft check." While the authors mitigated this, it underscores the inherent fragility of any evaluation pipeline that still relies on an LLM as a judge, even in a hybrid system.

**Questions:**

How might the results change with more sophisticated orchestration instead of just ReAct? Is the benchmark senstive to this?

Where do these sub-agents come from?

The paper mentions mitigating the observed reward hacking by adding a "style" check. Could the authors elaborate on this? Do they anticipate other, more subtle forms of hacking? What is the long-term strategy for ensuring verifier robustness as agents become more adept at gaming these systems?

---

> ### Author Response · Authors · 2025-11-19
> **Comment on Agent2Agent split and Reward-hacking**
>
> We thank Reviewer QMSH for their valuable feedback and questions. Their questions and feedback on the scaffold and the Agent2Agent split have been addressed in a global comment.
>
> ## On the “mobile-only” scope.
>
> “Foundational” in the abstract should refer to the ARE platform, not Gaia2; we will add the missing comma to fix this typo.
>
> **Reviewer QMSH asks whether Gaia2, implemented in a “mobile” environment, “would translate to other common agentic environments”.**
>
> This question aligns with the core intent of our contribution: ARE is designed to provide the foundations for building any agentic environment.
>
> Gaia2 is only one instantiation of ARE. The platform is environment-agnostic and defines an API-based framework supporting partial observability, asynchronicity, and multi-app workflows (Section 3). Gaia2 applies these abstractions to a consumer mobile environment, but the same structure extends naturally to domains such as desktop automation, customer support, and web browsing. The API tool for the Chats app would be the same on mobile or a desktop environment. Creating a new environment requires only defining new tools or apps since interfaces are the same; ARE then supplies the event-driven execution, instrumentation, and verification pipeline.
>
> To demonstrate this generality, we managed to reimplement several existing benchmarks on top of ARE (tau-bench, tau-bench^2, BFCL-v3, GAIA). Most do not take advantage of the richer capabilities ARE enables (e.g., temporal reasoning, multi-agent interactions), confirming that the platform both subsumes current benchmarks and provides a foundation for the next generation of agentic evaluations. We will clarify this scope distinction in the revision.
>
>
> ## On verifier robustness and reward-hacking.
>
> **Reviewer QMSH raises a concern about the long-term robustness of our verifier, citing the reward-hacking case in Appendix B.2.3.**
>
> We thank the reviewer for the question on long-term verifier robustness. We identified the reward-hacking issue in Appendix B.2.3 during our own RL training runs, and it was limited to the Search split because, by design, these tasks contain a single terminal write and thus only one judgment signal. After detecting this, we updated the verifier to ensure two independent judgment signals even in single-event scenarios. We subsequently ran dozens of new RL experiments to empirically stress-test the updated verifier and were unable to reproduce any exploitation. We will clarify this in the revision, as our iterative RL-based evaluation played a key role in strengthening the verifier’s reliability.
>
>
> The “style check” we added is a lightweight rubric that flags outputs containing artifacts or suspicious injections (e.g., template-breaking text intended to mislead the LLM judge). It was validated on a held-out set of 100 human-labeled outputs and closed the only loophole observed in the Search setting. More broadly, our strategy is to combine orthogonal constraints in ARE’s verifier (causal, semantic, syntactic) so that exploiting the system becomes costlier than solving the task. While LLM-based judging introduces some inherent fragility, it enables scalable evaluation and versatility, and ARE’s verifier abstraction is designed to evolve as agents improve.

---

> > ### Comment · Reviewer_QMSH · 2025-11-27
> >
> > Thank you for your response and clarifications. However, I remain concerned about the verifier: relying on LLM judges inherently invites reward hacking that heuristic "style checks" may not permanently solve, and the high cost/latency of such verifiers poses a significant barrier for practical, large-scale RL training. Additionally, while the single ReAct scaffold can be considered fair, it limits the exploration of broader model capabilities. Furthermore, the results of Parallel TC vs ReAct experiment for some models seem to show a notable gap. Recognizing the platform's solid contribution despite these unresolved limitations, I will maintain my score.

---

> > > ### Author Response · Authors · 2025-11-28
> > > **Response to reviewer QMSH**
> > >
> > > We thank reviewer QMSH for taking the time to go through our rebuttal and raising their concerns. To continue this interesting discussion, which might improve the paper's understanding for future readers, we would like to clarify a few points.
> > >
> > > **"LLM-judge can be hacked":** Gaia2 verifier is not solely an LLM judge. The verifier prioritizes programmatic checks and only falls back to the LLM judge for semantic verification when necessary (cf. line 281). It is also important to note that even benchmarks evaluated purely via exact match can be gamed—this is a generic property of metrics, not a flaw specific to LLM-based judging. We believe that the best way to mitigate this is to combine both, motivating our choice to use LLM as a Judge and exact-match validations in our verifier.
> > >
> > > **On "Verifier latency and cost":** We respectfully disagree that verifiers in their current form pose a barrier to training. The verifier runtime is negligible compared to agent rollout time, with only ~3% of total time spent on scoring and ~97% on trajectory generation. This explicitly demonstrates that the design supports scalable data generation for RLVR without the latency bottleneck the reviewer fears.
> > >
> > > A potential research direction is to increase the compute for the verifiers, as models will become more capable. In this context, it is possible that verifiers will become less scalable, but our current results show that we are not there yet, with a benchmark not saturated while leveraging verifiers 2 orders of magnitude less costly than the agent.
> > >
> > > **We both agree ReAct is a fair baseline but has limitations.** We would like to highlight that this is exactly the purpose for a benchmark: to emphasize limitations of current standards and encourage research in new state-of-the-art. We hope Gaia2 will help in that regard to go beyond ReAct.

---

### Official Review · Reviewer_cZHA · 2025-11-04

**Soundness:** 3
**Presentation:** 3
**Contribution:** 3
**Rating:** 8
**Confidence:** 4

**Summary:**

This paper introduces Gaia2, a novel benchmark for LLM agents, and the underlying platform Agents Research Environments (ARE) building it. This paper's contributions are threefold, aimed at advancing the evaluation of LLM agents in dynamic environments and realistic scenarios.:

- First, it open-sourced the ARE platform for building and running agentic evals. ARE is designed from the ground up for building asynchronous, event-driven benchmarks while also support scalable evaluation and data generation for Reinforcement Learning (RL).

- Second, the paper presents the Gaia2 benchmark. Gaia2 is the first benchmark to unify a set of critical, real-world agent capabilities under one evaluation, including asynchronous execution, temporal reasoning, robustness to environmental noise, ambiguity resolution, and multi-agent collaboration.

- Third, the authors conduct a comprehensive empirical study on Gaia2 using a suite of leading models. This study exposes fundamental trade-offs between agent reasoning strength, efficiency (latency), robustness, and operational cost.

**Strengths:**

- True Asynchronicity and Temporal Dynamics enabled by the async nature of the ARE platform and the event driven/DAG design for constructing more tass.

- A Scalable, Open-Source Platform (ARE) that allows the community to build new, more complex scenarios on top of this work, ensuring its long-term relevance.

- Well-Designed Verifier for RLVR: The new action-level verifier is proven to be effective, achieving much higher precision and recall than a LLM-only judge.

- The analysis of cost-performance-time trade-offs provides a practical, multi-dimensional view of agent performance that is highly relevant for real-world deployment.

- The benchmark is not saturated and successfully identifies clear weaknesses in current SOTA models. The low scores on Noise, Ambiguity, and Agent2Agent splits provide a clear roadmap for future research.

**Weaknesses:**

- The verifier is designed to check against a "minimal oracle sequence" of write actions, which might be brittle in some cases. It appears to evaluate path optimality rather than goal completion. An agent that makes a mistake, self-corrects (e.g., books the wrong cab, cancels, books the correct one), and ultimately reaches the correct state would be marked as a failure.

- The "inverse scaling" result might be confounded by the agent's orchestration. Given that all models use a single threaded simple ReAct-like scaffold, this might conflate model inference latency with scaffold inefficiency. A more advanced orchestration (e.g., parallel tool calls, or a planner that coordinate) might allow a "slow" model to succeed. And overall the extremely low score on this split seem more for consideration under specific circumstance rather than general capability limitation.

- The capability taxonomy is somewhat muddled, mixing action types (Execution, Search) with problem constraints (Time, Noise). While the design to stress test one capability in isolation is reasonable. Real-world tasks are almost never this clean; they require a composition of skills and the lack of a "Compositional" split is a weak point

**Questions:**

- Could the verifier be modified to support "goal completion" rather than "path optimality"? How would you propose handling valid, non-minimal trajectories that involve agent self-correction? Overall feel it might be helpful to better structure the categorization of the capability, and some break down/analysis on individual categories, e.g., does task complexity/length affect the performance on execution/search

- Given that the current scenarios test capabilities in isolation, why not adding compositional split or there are some tests showing it is not needed?

- Agent2Agent setting seem to be a bit different than the other settings, which can be used to evaluated both the main agents' planning and coordination skill, especially given partial observability of the action space, as well as the sub-agents ability on effectively communicate with the main agent and complete task, the experiments and design for this part could worth further exploration.

---

> ### Author Response · Authors · 2025-11-19
> **Comment on the Verifier and Capabilities**
>
> We thank Reviewer cZHA for their valuable feedback and questions. Their questions and feedback on the scaffold and the Agent2Agent split have been addressed in a global comment.
>
> ## On “goal oriented” vs “path optimal” Verifier
>
> **Reviewer cZHA asked why the verifier “evaluates path optimality rather than goal completion.”**
>
>  In Gaia2, read and write actions are explicitly separated: only write actions modify the user’s environment, and producing the correct write corresponds to achieving the scenario’s goal (see on line 258). Agents may take any sequence of reads to reach that goal, and the verifier does not penalize non-optimal reading paths. Because Gaia2 includes a notion of time, an invalid write action, even if later self-corrected, temporarily violates the environment state and is therefore irreversible. When temporal ordering matters, we specify it; otherwise, the verifier is agnostic: for example, sending two messages to two friends in any order is treated equivalently.
>
> While this is a design choice, the verifier abstractions in ARE are modular enough to create a goal-completion or tolerance-based version, which, for example, could only check the final state of the environment, allowing self-correction.
>
>
> ## On Capability Taxonomy
>
> **Reviewer cZHA notes that the current capability taxonomy appears to mix action types (Execution, Search) with constraints (Time, Noise).**
>
> We clarify the split decomposition: (i) core capabilities: Execution, Search, Ambiguity, Adaptability, and Time; and (ii) augmentation settings: Noise and Agent2Agent, which we can apply on any scenario thanks to the invariant nature of annotations (see line 208). This distinction is currently explained in Appendix B.1.4; we will move this clarification into the main text to avoid confusion.
>
>
> **The reviewer also stresses the value of stress-testing capabilities in isolation but asks why there is no explicit “compositional split”.**
>
> Gaia2’s splits enable clear failure-mode attribution (e.g., Table 2 shows Grok-4 performs well on Search but underperforms elsewhere). In practice, any natural task is inherently compositional and touches multiple capability categories. Our splits ensure that, for each category (e.g., Search), every scenario requires that capability to succeed, while still involving others. This design produces realistic, multi-skill tasks while still uncovering systematic capability biases in models, such as Grok-4’s strong Search behavior. We tested scenarios explicitly combining more than 3–4 capabilities, but annotators found them unnatural and too complex to yield a clean evaluation signal, so we chose not to introduce an additional “compositional” split.
>
>
> **Reviewer cZHA asks for a breakdown of task complexity and whether factors such as scenario length affect Execution/Search performance.**
>
> As noted in Section 4.2 (line 251), complexity is difficult to quantify: oracle write counts are poor predictors (e.g., multi-write tasks may be easy, while single-write Search tasks can require extensive exploration), and read actions are unannotated by design. Although using the number of oracle events is tempting, our annotation experience showed it does not correlate well with actual difficulty. We use a more empirical approach for calibration: running models on scenarios. We strongly think that what is difficult for humans is not necessarily hard for LLMs. To better understand what drives model performance, we also provide behavioral proxies (steps, tool calls, tokens; Fig. 18 in the Appendix), which correlate more strongly with true task difficulty.

---

### Author Response · Authors · 2025-11-19
**General Comment**

This global response addresses questions raised by multiple reviewers; reviewer-specific points are handled in their respective sections. We sincerely thank all reviewers for their constructive feedback and will release a revised version soon.


## On the ReAct scaffolding

**Reviewers cZHA and QMSH both question whether the ReAct scaffold might act as a bottleneck, potentially limiting the model’s intrinsic capabilities in the Gaia2 benchmark.**

We asked ourselves the same question, and decided to choose ReAct intentionally which we discuss in the Experimental Setup (line 287), motivated by its model agnostic nature and its ability to support both closed and OSS models. We think it is a fair evaluation setting.

Following reviewer cZHA’s suggestion, we ran a Parallel ToolCalling (PTC) scaffold ablation (see table). PTC lowers wall-clock latency (e.g., GPT-5 Exec 610→175 s; Claude Exec 205→137 s) with only modest deltas (-1.3 to 1.5 pp on Time), all within measured variance (Table 2). Model ranking is unchanged, confirming that our qualitative conclusions are not artifacts of the scaffold. Full results will be added to the revised appendix.

As noted in our Discussion (line 480), we agree that better orchestration for async environments is needed. This motivated the Time category, designed to expose sequential-reasoning limits independent of the scaffold and to accelerate research on more general orchestrations.



**Table**: Ablations of 3 models with Parallel TC vs ReAct scaffold. Δ = Parallel TC - ReAct = net contribution of PTC over ReAct.

| Model            | Split      | ReAct pass@1 | Parallel TC pass@1 | Δ pass@1 (pp) | Δ avg time (s)           | Δ avg steps        | Δ avg output tokens        |
|------------------|------------|--------------|----------------------|----------------|------------------------|----------------------|------------------------------|
| Llama Maverick   | Execution  | 13.8         | 7.5                  | -6.3           | 205 - 134 = **+71**    | 16.3 - 17.3 = **-1** | 5946 - 4160 = **+1786**     |
| Llama Maverick   | Time       | 1.2          | 2.0                  | +0.8           | 129 - 132 = **-3**     | 16.1 - 17.2 = **-1.1** | 5390 - 3150 = **+2240**     |
| Claude 4 Sonnet  | Execution  | 57.9         | 59.7                 | +1.8           | 137 - 205 = **-68**    | 16.3 - 27 = **-10.7** | 3220 - 3565 = **-345**      |
| Claude 4 Sonnet  | Time       | 8.1          | 9.5                  | +1.4           | 192 - 200 = **-8**     | 22.2 - 24.6 = **-2.4** | 3183 - 3150 = **+33**       |
| GPT-5 (minimal)  | Execution  | 31.9         | 34.9                 | +3.0           | 136 - 200 = **-64**    | 14 - 28 = **-14**     | 3906 - 4066 = **-160**      |
| GPT-5 (minimal)  | Time       | 5.2          | 6.7                  | +1.5           | 180 - 157 = **+23**    | 21 - 21 = **0**       | 3638 - 2608 = **+1030**     |
| GPT-5 (low)      | Execution  | 52.7         | 51.7                 | -1.0           | 175 - 610 = **-435**   | 10 - 23 = **-13**     | 6439 - 11548 = **-5109**    |
| GPT-5 (low)      | Time       | 2.3          | 1.0                  | -1.3           | 197 - 404 = **-207**   | 11.9 - 13.8 = **-1.9** | 2971 - 7396 = **-4425**     |


## On the Agent2Agent Split

**Reviewer cZHA and QSMH express their interest in the Agent2Agent (A2A) split and request clearer details on how app-agents operate and what capabilities this setting probes.**
Each app is replaced by an app-agent with access only to that app’s tools (see on line 213). The main agent must delegate subtasks (e.g. ask_calendar_expert(query)) to interact with apps, and app-agents only execute the delegated query and return a final report. They are invoked on demand and do not act autonomously. In our evaluation, app-agents use the same model as the main agent, though heterogeneous configurations are supported. This setting tests the main agent’s ability to plan and coordinate under partial observability, while app-agents must reliably execute and communicate. This will be clarified in the revised version.

**Reviewer cZHA noted the value of this setup for studying planning, coordination, and sub-agent communication.** We include an additional A2A ablation:

| **Main-Agent**           | **App-Agent**            | **pass@1**   |
|--------------------------|---------------------------|--------------|
| Llama 4 Maverick         | Llama 4 Maverick          | 8.5 ± 1.7    |
| Llama 4 Maverick         | Claude 4 Sonnet           | 16.2 ± 0.7   |
| Claude 4 Sonnet          | Llama 4 Maverick          | 18.3 ± 0.7   |
| Claude 4 Sonnet          | Claude 4 Sonnet           | 29.3 ± 2.9   |

These results show that Gaia2 A2A split jointly stresses planning capabilities and reliable execution and that heterogeneous multi-agent teams can outperform homogeneous ones when replacing weaker models. Gaia2 and ARE provide a good starting point to bootstrap research on multi-agent systems.

---

> ### Author Response · Authors · 2025-11-25
> **Submission of Revised Paper & Summary of Updates**
>
> We thank the reviewers for their constructive feedback and the opportunity to strengthen our work. Following our comments, we have uploaded a revised version with the following changelog:
>
> - **New experiment: Orchestration Analysis (ReAct vs. Parallel Tool Calling), Appendix B.3.2.**
>   We have added a detailed ablation study comparing ReAct against a Parallel ToolCalling (PTC) scaffold. We show that while PTC can improve latency/token usage, the model's performance remain the same, confirming that our **ReAct baseline is fair** and not a bottleneck for performance.
>
> - **New experiment: Agent2Agent (A2A) Expansion, Table 3.**
>   We provide an additional ablation showing that heterogeneous teams (pairing different models as main agents and sub-agents) can outperform homogeneous ones. We also clarified the lifecycle and communication protocol of app-agents in the main text.
>
> - **Clarification on ARE’s Foundational Scope, Section 3.**
>   We have revised the ARE section and abstract to clearly distinguish that ARE is the foundational, environment-agnostic platform (supporting benchmarks like BFCL-v3 and Gaia), while Gaia2 is a specific mobile instantiation. We further explain how these concepts transfer to other setups like web navigation.
>
> - **Revised Capability Taxonomy, Section 4.1.**
>   We clarified the distinction between "Core Capabilities" (e.g., Search, Execution) and "Augmentations" (e.g., Noise, A2A) by moving this definition to the main text to avoid confusion regarding the split design.
>
> - **Analysis on Performance Drivers, Figure 7.**
>   We have added an analysis in the main body regarding performance drivers, using behavioral proxies (steps, tokens) to better quantify task difficulty and model behavior.
>
> - **Smaller improvements.**
>   We made several smaller improvements, such as fixing the "foundational" attribution in the abstract and expanding the discussion on the need for future research in asynchronous orchestration.
>
> We believe these updates address the core concerns raised by reviewers and discussed in the previous comments, and strengthen the pape. We look forward to any further discussion.

---

### Meta-Review · Area_Chair_Du6z · 2026-01-07

**Summary:**

This paper proposes GAIA2, a successor of GAIA but in a diverse simulated environments, where the agents can access many tools created by the authors. Reviewers agree this is a strong benchmark/platform contribution: GAIA2 targets asynchronous, dynamic agent environments with action-level verification and fine-grained capability splits, and the accompanying ARE framework is seen as a meaningful framework for future work for potential data synthesis and RL. The main concerns raised in reviews/discussion are not about “whether the benchmark is useful,” but about how to interpret the results and how robust/general the setup is: (i) whether the chosen single-threaded ReAct-style orchestration bottlenecks models and confounds “inverse scaling” / time-sensitive failures, (ii) whether the write-action verifier is overly “path-optimal” and can penalize self-correction, (iii) clarity and interpretability of the Agent2Agent setting and capability taxonomy (and whether a compositional split is needed).

Overall, I think GAIA2 presents a strong benchmark and a useful environment ARE. Thus I recommend a clear acceptance of this paper.

**Reviewer Concerns:**

Concerns addressed in rebuttal:
1. Orchestration confound: authors ran an ablation with Parallel Tool Calling vs ReAct and argue rankings and main conclusions are stable, suggesting ReAct is not the main bottleneck.
2. Agent2Agent clarity
3. Verifier goal-completion vs path-optimality: authors explained why incorrect writes can be irreversible under time, and noted ARE’s modular verifier could support tolerance/final-state variants.

Concerns still outstanding:
1. Reliance on any LLM-judge component is viewed as inherently gameable long-term

**Reviewer Scores:**

The reviewers are already very positive about this paper with scores 8,6,10, I think they would maintain the positive ratings.

---

### Decision · Program_Chairs · 2026-01-26

Accept (Oral)